# Active Sequential Two-Sample Testing

**Weizhi Li**[*]                                                                          *weizhili@asu.edu*
*Arizona State University*
*Los Alamos National Laboratory*

**Prad Kadambi**                                                                        *pkadambi@asu.edu*
*Arizona State University*

**Pouria Saidi**                                                                          *psaidi@asu.edu*
*Arizona State University*

**Karthikeyan Natesan Ramamurthy**                                        *knatesa@us.ibm.com*
*IBM Research*

**Gautam Dasarathy**                                                                  *gautamd@asu.edu*
*Arizona State University*

**Visar Berisha**                                                                          *visar@asu.edu*
*Arizona State University*

**Reviewed on OpenReview:** *https://openreview.net/forum?id=EzPRgIq2Tk*

## Abstract

A two-sample hypothesis test is a statistical procedure used to determine whether the distributions generating two samples are identical. We consider the two-sample testing problem in a new scenario where the sample measurements (or sample features) are inexpensive to access, but their group memberships (or labels) are costly. To address the problem, we devise the first *active sequential two-sample testing framework* that not only sequentially but also *actively queries*. Our test statistic is a likelihood ratio where one likelihood is found by maximization over all class priors, and the other is provided by a probabilistic classification model. The classification model is adaptively updated and used to predict where the (unlabelled) features have a high dependency on labels; labeling the "high-dependency" features leads to the increased power of the proposed testing framework. In theory, we provide the proof that our framework produces an *anytime-valid p*-value. In addition, we characterize the proposed framework's gain in testing power by analyzing the mutual information between the feature and label variables in asymptotic and finite-sample scenarios. In practice, we introduce an instantiation of our framework and evaluate it using several experiments; the experiments on the synthetic, MNIST, and application-specific datasets demonstrate that the testing power of the instantiated active sequential test significantly increases while the Type I error is under control.

## 1 Introduction

The two-sample test is a statistical hypothesis test applied to data samples (or measurements) from two distributions. The goal is to test if the data supports the hypothesis that the distributions are different. If we consider each data point as a feature and label (which tells us which distribution the data is from) pair, then the two-sample test is equivalent to the problem of testing the dependence between the features and the labels. Viewed with this lens, the null hypothesis for the two-sample test states that the feature and

---

[*]Work done when the author was at Arizona State University.

label variables are independent, and the alternate hypothesis states the opposite. The analyst performing the two-sample test needs to decide between the null and the alternative hypotheses with data from the two distributions.

The analyst typically knows little about the difficulty of a two-sample testing problem before running the test. Fixing the sample size a priori may result in a test that needs to collect additional evidence to arrive at a final decision (if the problem is hard) or in an inefficient test with over-collected data (if the problem is simple). To address this dichotomy, the research community has proposed sequential two-sample tests (Wald, 1992; Lhéritier & Cazals, 2018; Hajnal, 1961; Shekhar & Ramdas, 2021; Balsubramani & Ramdas, 2015) that allow the analyst to sequentially collect data and monitor statistical evidence, i.e., a statistic is computed from the data. The test can stop anytime when sufficient evidence has been accumulated to make a decision.

Existing sequential two-sample tests (Wald, 1992; Lhéritier & Cazals, 2018; Hajnal, 1961; Shekhar & Ramdas, 2021; Balsubramani & Ramdas, 2015) are devised to collect both sample features and sample labels simultaneously. **In this paper, we consider the problem of sequential two-sample testing in a novel and practical setting where the cost of obtaining sample labels is high, but accessing sample features is inexpensive**. As a result, the analyst can obtain a large collection of sample features without labels; she will need to sequentially query the label of the sample features in the collection to perform the two-sample testing while ensuring the query complexity (i.e., the number of queried labels) doesn't exceed a label budget. A motivation for this formulation comes from the field of digital health: Physicians seek inexpensive digital measurements (e.g., gait, speech, typing speed measured using a patient's smartphone) to replace traditional biomarkers (e.g., the amyloid buildup that indicates Alzheimer's progression) which are often costly to access; hence they need to validate the dependency between the digital measurements (feature variables) and traditional biomarkers (label variables). While validation studies can access large registries to collect digital measurements remotely at scale, there is a fixed label budget for the expensive biomarker measures. An efficient sequential design would reveal the dependency between the features and the labels using only a *reasonable* label budget.

In this paper, we propose the active sequential testing framework shown in Figure 1. The framework initializes a classifier to model probabilities of sample labels given features using an initial random sample; next, depending on the classifier's outputs, the framework queries the labels of features predicted to have a high dependency with the labels and constructs a test statistic $w$. The framework rejects the null if $w$ is smaller than a pre-defined significance level $\alpha$; otherwise, the framework stops and retains the null if the label budget runs out or re-enters the label query and decision-making, enabling a sequential testing process.

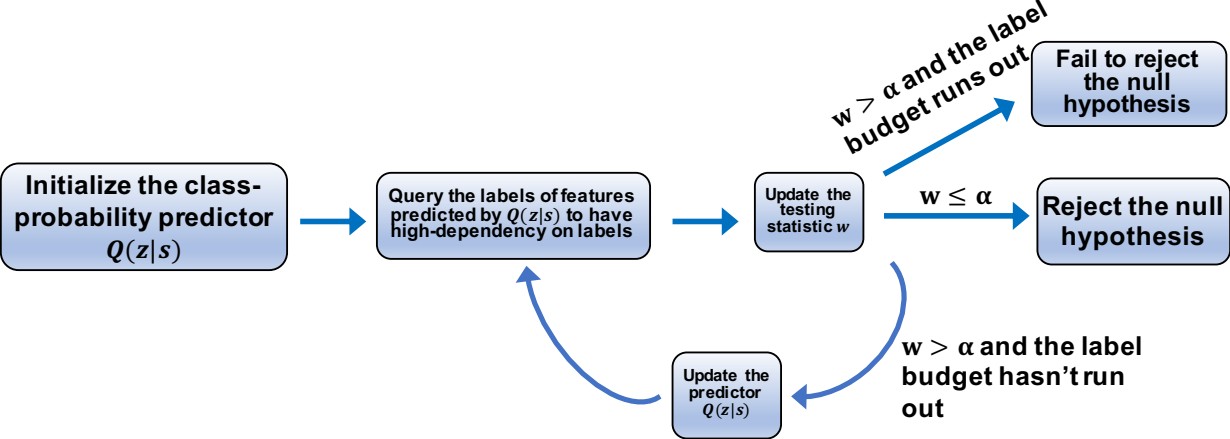

Figure 1: The active sequential two-sample testing framework.

The test statistic $w$ in the framework is based on the likelihood ratio between the likelihood constructed under the null that feature and label variables are independent and the likelihood constructed under the alternative

that the dependency between the feature and label variables exists. Such a likelihood ratio two-sample test statistic has been first proposed in (Lhéritier & Cazals, 2018) to develop a non-active sequential two-sample test capable of controlling the Type I error (i.e., the probability of a decision made on the alternative when the null is true). We adapt the original test statistic by replacing the pre-defined label probability prior with a maximum likelihood estimate to satisfy our considered setting that the label prior is unknown. More importantly, our framework actively labels the features that are predicted to have a high dependency on labels. We will characterize the benefits of the active query over the random query by the change of mutual information between feature and label variables in the asymptotic and finite-sample scenarios. In practice, we suggest using an active query scheme called bimodal query proposed in (Li et al., 2022), in which the scheme labels samples with the highest class one or zero probabilities.

We summarize the main contributions of our work as follows:

- We introduce the first *active sequential two-sample testing framework.* We prove that the proposed framework produces an *anytime-valid p*-value to achieve Type I error control. Furthermore, we provide an information-theoretic interpretation of the proposed framework. We prove that, asymptotically, the framework is capable of generating the largest mutual information (MI) between feature and label variables under standard conditions (Györfi et al., 2002); and we also analyze the gain of the testing power for the proposed framework over its passive query parallel in the finite-sample scenario through MI.

- We instantiate the framework using the bimodal query (Li et al., 2022) (i.e., queries the labels of the samples that have the highest class one or zero probabilities) as the label query scheme. We perform extensive experiments on synthetic data, MNIST, and an application-specific Alzheimer's disease dataset to demonstrate the effectiveness of the instantiated framework. Our proposed test exhibits a significant reduction of the Type II error using fewer labeled samples compared with a non-active sequential testing baseline.

## 2 Related Works

The author of (Student, 1908) developed the *t*-test, probably the simplest form of a two-sample test that compares the mean difference of two samples of uni-variate data. Since then, the research community has expanded the two-sample test to many other forms, e.g., the hotelling test (Hotelling, 1992), the Friedman-Rafsky test (Friedman & Rafsky, 1979), the kernel two-sample test (Gretton et al., 2012) and the classifier two-sample test (Lopez-Paz & Oquab, 2016) for the multi-variate case. These tests are constructed with various statistics, including the Mahalanobis distance, the measurement over a graph, a kernel embedding, or classifier accuracy, all in service of increasing testing power while controlling the Type I error. In particular, (Friedman & Rafsky, 1979; Gretton et al., 2012; Lopez-Paz & Oquab, 2016) test if the data from two samples is distributionally different, which is a generalization of the hotelling and $t-$test (Student, 1908; Hotelling, 1992) that only detect the mean difference of two samples. These two-sample tests are batch tests that have been extensively used subject to a fixed-sample size: When the collection of experimental data ends, an analyst performs the two-sample tests on the data and makes a decision; she is not allowed to continue to collect and incorporate more data into the testing after a decision made, as that will inflate the Type I error.

In contrast to the batch two-sample tests, the research community has developed a class of sequential two-sample tests (Lhéritier & Cazals, 2018; Shekhar & Ramdas, 2021; Pandeva et al., 2022) that allow the analyst to sequentially collect data and perform the two-sample test, enabling sequential decision-making. These sequential tests rectify the inflated Type I that will happen in the batch test with different statistical techniques such as Bonferroni correction (Dunn, 1961) and Ville's maximal inequality (Doob, 1939).

There are also several works that consider the active setting in two-sample testing. The authors of (Li et al., 2022) proposed a batch two-sample test combined with active learning when curated labeled data is unavailable and querying the data labels is expensive. Several studies have also considered sequential testing for developing active sequential hypothesis tests (Naghshvar & Javidi, 2013; Chernoff, 1959; Bessler, 1960; Blot & Meeter, 1973; Keener, 1984; Kiefer & Sacks, 1963). However, these tests require a clear parametric description of the statistical models of the hypotheses. The authors of (Duan et al., 2022) developed an

interactive rank test, which is distribution-free and can similarly perform the sequential two-sample testing in the active learning setting.

The work proposed herein uses the label query scheme in (Li et al., 2022) to develop the first multivariate non-parametric sequential test for the active learning setting with a novel test statistic and theoretical results. We demonstrate that the test controls the Type I error via Ville's maximal inequality (See Theorem 5.1). Ville's maximal inequality results in higher testing power than the Bonferroni correction for sequential testing (Shekhar & Ramdas, 2021; Ramdas et al., 2022).

While our framework in Figure 1 employs the label query scheme introduced in (Li et al., 2022), it offers distinct advantages over (Li et al., 2022):

- Our proposed framework follows a sequential design. Upon accumulating sufficient evidence to reject the null hypothesis, our design automatically stops label collection before exhausting the label budget. In contrast, the batch design in (Li et al., 2022) invariably exhausts the label budget.

- Utilizing a different test statistic, our framework enables finite-sample analysis, which is not provided in (Li et al., 2022).

## 3 Problem Statement and Preliminaries

### 3.1 Notations

We use a pair of random variables $(\mathbf{S}, Z)$ to denote a feature and its label variables whose realization is $(\mathbf{s}, z) \in \mathbb{R}^d \times \{0, 1\}$. The variable pair $(\mathbf{S}, Z)$ admits a joint distribution $p_{\mathbf{S}Z}(\mathbf{s}, z)$. Furthermore, we write $\mathcal{S}$ to denote the support of $p_{\mathbf{S}}(\mathbf{s})$. Formally, a two-sample testing problem consists of null hypothesis $H_0$ that states $p_{\mathbf{S}|Z=0}(\mathbf{s}) = p_{\mathbf{S}|Z=1}(\mathbf{s})$ and an alternative hypothesis $H_1$ that states $p_{\mathbf{S}|Z=0}(s) \neq p_{\mathbf{S}|Z=1}(s)$. An analyst collects a sequence $((\mathbf{s}, z)_i)_{i=1}^N$ of $N$ realizations of $(\mathbf{S}, Z)$ to test $H_0$ against $H_1$. The problem is equivalent to testing the independency between $\mathbf{S}$ and $Z$. Therefore, we equivalently restate the hypothesis test as follows:

$$H_0 : p_{\mathbf{S}Z}(\mathbf{s}, z) = p_{\mathbf{S}}(\mathbf{s})P_Z(z), \forall \mathbf{s} \in \mathcal{S}$$
$$H_1 : p_{\mathbf{S}Z}(\mathbf{s}, z) \neq p_{\mathbf{S}}(\mathbf{s})P_Z(z), \exists \mathbf{s} \in \mathcal{S} \tag{1}$$

Moving forward, we omit the subscripts in $p_{\mathbf{S}Z}(\mathbf{s}, z)$, $P_Z(z)$ and $p_{\mathbf{S}}(\mathbf{s})$ and write them as $p(\mathbf{s}, z)$, $P(z)$ and $p(\mathbf{s})$. In addition, we use $\mathbf{s}^N$, $z^N$ and $(\mathbf{s}, z)^N$ to denote sequences of samples $(\mathbf{s}_i)_{i=1}^N$, $(z_i)_{i=1}^N$ and $((\mathbf{s}, z)_i)_{i=1}^N$ respectively. We use similar notation throughout the paper.

### 3.2 The problem

In the typical setting of a sequential two-sample test, an analyst does not have prior knowledge of sample features. The analyst sequentially collects both sample features and their labels *simultaneously* with the corresponding random variable pair $(\mathbf{S}, Z)$ i.i.d. generated from a data-generating process, i.e., $p(\mathbf{s}, z)$. We consider a variant of the setting in which accessing sample features is free/inexpensive. Consequently, the analyst collects a large set $\mathcal{S}_u$ of sample features before performing a sequential test. However, accessing the label of a feature in $\mathcal{S}_u$ is costly. **We assume the following fact throughout the paper**: The already-collected $\mathcal{S}_u$ is the result of a sample feature collection process where all $\mathbf{s}_i \in \mathcal{S}_u$ are realizations of random variables $\mathbf{S}_i$ i.i.d. generated from $p(\mathbf{s})$. There exists an oracle to return a label $z_i$ of $\mathbf{s}_i \in \mathcal{S}_u$ with the corresponding random variable $Z_i$ and $\mathbf{S}_i$ admitting the posterior probability $P(z_i|\mathbf{s}_i)$. We consider the following *new* sequential two-sample testing problem:

> **An active sequential two-sample testing problem**: Suppose $\mathcal{S}_u$ is an unlabeled feature set, there exists an oracle to return a label $z$ of $\mathbf{s} \in \mathcal{S}_u$, and $N_q$ is a limit on the number of times the oracle can be queried (e.g., the label budget). An analyst sequentially queries the oracle for the $z$ of $\mathbf{s} \in \mathcal{S}_u$. After querying a new $z_n$ of $\mathbf{s}_n$, for $1 \leq n \leq N_q$, the analyst needs to decide whether to terminate the label querying process and make a decision (i.e., whether to reject $H_0$) or continue with the querying process if $n < N_q$.

An analyst actively labeling $\mathbf{s}_n \in \mathcal{S}_u$ may result in non-i.i.d pairs of $(\mathbf{S}, Z)$; hence the distribution of $(\mathbf{S}, Z)$ is shifted away from $p(\mathbf{s}, z)$. In contrast, an analyst passively (or randomly) labeling $\mathbf{s}_n \in \mathcal{S}_u$ maintains $(\mathbf{S}, Z) \sim p(\mathbf{s}, z)$.

### 3.3 Evaluation metrics for the problem

In the following, we introduce the evaluation metrics used throughout the paper.

- **Type I error** $P_0$: The probability of rejecting $H_0$ when $H_0$ is true.

- **Type II error** $P_1$: The probability of rejecting $H_1$ when $H_1$ is true.

- **Testing power**: The probability of rejecting $H_0$ when $H_1$ is true. In other words, Testing power $= 1 - P_1$.

Testing power and Type II error are interchangeably used in the methodology and experiment sections (Section 5 and 6).

### 3.4 Attributes of an active two-sample test

As already generalized in many two-sample testing literature such as (Johari et al., 2022; Wald, 1992; Lhéritier & Cazals, 2018; Shekhar & Ramdas, 2021; Welch, 1990), a conventional procedure for sequential two-sample testing is to compute a $p$-value from sequentially observed samples and compare it to a pre-defined significance level $\alpha \in [0, 1]$ anytime. The analyst rejects $H_0$ and stops the testing if $p \leq \alpha$. For more details, see (Wasserstein & Lazar, 2016). In addition, as the test proposed in what follows is endowed with active querying to reduce the number of label queries, the active sequential test is anticipated to spend fewer labels than a passive (random-query) test to reject $H_1$ when $H_1$ is true. In summary, an active sequential two-sample test has the following four attributes:

- The test generates an *anytime-valid* $p$-value such that $P_0(p \leq \alpha) \leq \alpha$ holds at anytime of the sequential testing process. $P_0$ is exactly the *Type I error* and that implies the Type I is upper-bounded by $\alpha$.

- The test has a high *testing power* $P_1(p \leq \alpha)$.

- The test is *consistent* such that $P_1(p \leq \alpha) = 1$ under $H_1$ when the test sample size goes to infinity.

- The test has higher $P_1$ than the passive test given the same label budgets.

## 4  A Sequential Two-Sample Testing Statistic

We follow the well-known likelihood ratio test (Wilks, 1938) to construct a sequential testing statistic. We use the statistical models that characterize the label generation processes conditional on the observed sample features under $H_0$ and $H_1$. More precisely, under $H_0$, we have $P(z|\mathbf{s}) = P(z), \forall \mathbf{s} \in \mathcal{S}$; that is, when $S$ and $Z$ are independent, the posterior probability $P(z|\mathbf{s})$ is the same for any $\mathbf{s}$ in the support $\mathcal{S}$ of $p(\mathbf{s})$. In contrast, under $H_1$, we have the following statistical model: $\exists s \in \mathcal{S}, P(z|\mathbf{s}) \neq P(z)$. We sequentially collect sample data $(\mathbf{s}, z)$, and when a new observation $(\mathbf{s}_n, z_n)$ arrives, we construct a likelihood ratio $w_n$: With $w_0 = 1$, $w_n = w_{n-1} \frac{P(z_n)}{P(z_n|\mathbf{s}_n)} = \prod_{i=1}^n \frac{P(z_i)}{P(z_i|\mathbf{s}_i)}, n \geq 1$ to assess $H_0$ against $H_1$.

The statistical models $P(z)$ and $P(z|\mathbf{s})$ are unknown. To formulate our two-sample test, we will use a likelihood estimate $\hat{P}(z^n)$ that is maximized over all the class priors to replace $P(z^n)$–the product of the class prior. In addition, we build a class-probability predictor $Q_n(z \mid \mathbf{s})$ with the past observed sample sequence $(\mathbf{s}, z)^{n-1}$ to model $P(z_n \mid \mathbf{s}_n)$–the posterior probability of $z_n$ given newly observed $\mathbf{s}_n$; any probabilistic classifier, such as a neural network and logistic function, can be used to build $Q_n(z \mid \mathbf{s})$. Additionally, $Q_1(z \mid \mathbf{s})$ indicates an initialized class-probability predictor[1]. We formally present our sequential testing statistic in the following:

---

**A sequential two-sample testing statistic**: Considering $(\mathbf{s}, z)$ is sequentially observed, and as a new $(\mathbf{s}_n, z_n)$ arrives, then, for $n = 1, 2, \cdots$, an analyst constructs

$$w_n = \frac{\hat{P}(z^n)}{Q(z^n \mid \mathbf{s}^n)} = \prod_{i=1}^{n} \frac{\hat{P}(z_i)}{Q_i(z_i \mid \mathbf{s}_i)} \qquad (2)$$

where $\hat{P}(Z = 1) = \frac{\sum_{i=1}^{n} z_i}{n}$ is a class prior chosen to maximize $\hat{P}(z^n)$ and $Q_i(z_i \mid \mathbf{s}_i)$ is the output of a class-probability predictor built by the past observed sequence $(\mathbf{s}, z)^{i-1}$.

---

We accordingly use $W_n$ to indicate a random variable of which $w_n$ is a realization. Our test statistic in equation 2 is a generalization of the test statistic proposed in (Lhéritier & Cazals, 2018). In contrast to that work, our test statistic does not require the prior class to be known. The analyst compares $w_n$ with $\alpha$ at every step $n$ starting from $n = 1$, stopping the test once encountering a step with $w_n \leq \alpha$. As a result, a small $w_n$ is favored under $H_1$ to reject $H_0$ for increasing testing power.

---

**Algorithm 1** Bimodal Query Based Active Sequential Two-Sample Testing (BQ-AST)

---
1: **Input:** $\mathcal{S}_u, \mathcal{A}, N_0, N_q, \alpha$
2: **Output:** Reject or fail to reject $H_0$
3: **Initialization**: Initialize $Q_1(z \mid \mathbf{s})$ using $\mathcal{A}$ with $N_0$ features uniformly sampled from $\mathcal{S}_u$ without replacement and then labeled.
4: **Active Sequential testing**:
5: **for** $n = 1$ to $N_q - N_0$ **do**
6:     Sample a feature $\mathbf{s}_n = \mathbf{s}_{q_0}$ or $\mathbf{s}_{q_1}$ with **fair chance** where $\mathbf{s}_{q_0} = \arg\max_{\mathbf{s}} \left[ Q_n(Z = 0|\mathbf{s}) \right], \forall \mathbf{s} \in \mathcal{S}_u$ and $\mathbf{s}_{q_1} = \arg\max_{\mathbf{s}} \left[ Q_n(Z = 1|\mathbf{s}) \right], \forall \mathbf{s} \in \mathcal{S}_u$
7:     Query the label $z_n$ of $\mathbf{s}_n$
8:     Update $w_n$ in equation 3 with $(\mathbf{s}_n, z_n)$ and $Q_n(z_n \mid \mathbf{s}_n)$
9:     **if** $w_n \leq \alpha$ **then**
10:         **Return** Reject $H_0$
11:     **else**
12:         **Update** $Q_n(z \mid \mathbf{s})$ with newly queried $(\mathbf{s}_n, z_n)$ and past training examples.
13:     **end if**
14: **end for**
15: **Return** Retain $H_0$

---

## 5 Active Sequential Two-Sample Testing

This section introduces the active sequential two-sample testing framework and its instantiation. We demonstrate that the framework produces an *anytime-valid* $p$-value regardless of the selected query scheme. We also provide the asymptotic and finite-sample performance of the framework with the testing power gain measured by the change of the mutual information between feature and label variables.

---

[1]It is possible to set $Q_1(z|\mathbf{s})$ as a random guess class-probability predictor, and then sequentially gather $(\mathbf{s}, z)$ for training; however, this would hurt the testing power. As suggested by Duan et al. (2022); Lhéritier & Cazals (2018), we initialize $Q_1(z|\mathbf{s})$ with a small set of samples randomly labeled and start the sequential testing after that.

### 5.1 An active sequential two-sample testing framework

A flow chart of the proposed framework is shown in Figure 1. Our framework starts by initializing the class-probability predictor $Q_n(z \mid \mathbf{s})$ at $n = 1$ with a small set of sample features randomly selected from $\mathcal{S}_u$ and then labeled. Then, the framework enters the sequential testing stage that iteratively performs the following: selects features in $\mathcal{S}_u$ predicted by $Q_n$ to have a high dependency on their labels, update the statistic $w_n$, decide whether we can reject $H_0$ and update $Q_n$ if the test has not stopped. We formally introduce our active sequential two-sample testing framework as follows,

> **An active sequential two-sample testing framework**: Suppose $N_q$ is a label budget and $\alpha$ is a significance level. An analyst uses the proposed framework to sequentially and actively query the label $z_n$ of $\mathbf{s}_n$ from an unlabelled feature set $\mathcal{S}_u$ based on the predictions of $Q_n(z \mid \mathbf{s})$. As a new $z_n$ of $\mathbf{s}_n$ is queried, the analyst constructs the following statistic
>
> $$w_n = \prod_{i=1}^{n} \frac{\hat{P}(z_i)}{Q_i(z_i \mid \mathbf{s}_i)}. \tag{3}$$
>
> The analyst evaluates $w_n$ and makes one of the following decisions: (1) rejects $H_0$ if $w_n \leq \alpha$; (2) retains $H_0$ if the label budget $N_q$ is exhausted and (1) is not satisfied; and (3) continues the test and updates $Q_n$ to $Q_{n+1}$ if (1) and (2) are not satisfied.

**Framework instantiation:** We provide a framework instantiation called **bimodal query based active sequential two-sample testing (BQ-AST)** described in Algorithm 1. The algorithm takes the following input: an unlabelled feature set $\mathcal{S}_u$, a probabilistic classification algorithm $\mathcal{A}$, the size $N_0$ of an initialization set used for $\mathcal{A}$, a label budget $N_q$ and a significance level $\alpha$. Then, the algorithm initializes a class-probability predictor $Q$ using $\mathcal{A}$ with a small set of randomly labeled samples. In the sequential testing stage, the algorithm uses bimodal query from Li et al. (2022) to sample $\mathbf{s}_n$ with samples having the highest posteriors from either class (e.g. a fair chance to select the highest $Q_n(Z = 0 \mid \mathbf{s})$ or $Q_n(Z = 1 \mid \mathbf{s})$) from $\mathcal{S}_u$, queries its label $z_n$ and updates the statistic $w_n$. Next, the algorithm compares $w_n$ with $\alpha$, and if $H_0$ is not rejected, update $Q_n$ with $(\mathbf{s}_n, z_n)$ and then re-enter the query labeling. The algorithm rejects $H_0$ if $w_n \leq \alpha$ or fails to reject $H_0$ if the label budget is exhausted.

The label budget $N_q$ in Algorithm 1 contains the labels for both initializing $Q_1(z \mid \mathbf{s})$ and constructing the statistic $w_n$. In what follows in this section, we simply use $N_q$ to denote the "label budget" allowed to be used after the initialization.

### 5.2 The proposed framework results in an anytime-valid $p$-value

Our framework rejects $H_0$ if the statistic $w_n \leq \alpha$. The following theorem states that under $H_0$, $w_n$ is an *anytime-valid $p$*-value.

**Theorem 5.1.** *If an analyst uses the proposed framework to sequentially query the oracle for $Z$ with $\mathbf{S} \in \mathcal{S}_u$ resulting in $(\mathbf{S}, Z)^n$, then we have the following under $H_0$,*

$$P_0\left(\exists n \in [N_q], W_n = \prod_{i=1}^{n} \frac{\hat{P}(Z_i)}{Q_i(Z_i \mid \mathbf{S}_i)} \leq \alpha\right) \leq \alpha \tag{4}$$

*where $N_q$ is a label budget and $\alpha$ is the pre-specified significance level.*

Theorem 5.1 implies the probability $P_0$ (or Type I error) that our framework mistakenly rejects $H_0$ is upper-bounded by $\alpha$. Briefly, we prove this by observing that the sequence $\left(\frac{1}{W_1}, \cdots, \frac{1}{W_n}\right)$ is upper-bounded by a martingale, and hence we use Ville's maximal inequality Durrett (2019); Doob (1939) to develop Theorem 5.1. See the Appendix for the complete proof.

### 5.3 Asymptotic properties of the proposed framework

This section provides the theoretical conditions under which the proposed framework asymptotically generates the smallest normalized statistic (normalization of the statistic in equation 3), or equivalently, maximally increases the mutual information between $\mathbf{S}$ and $Z$. Before that, we first define the *consistent bimodal query* as follows,

**Definition 5.2.** (*Consistent bimodal query*) Let $\mathcal{S}$ be the support of $p(\mathbf{s})$ that sample features are collected from and added to an unlabeled set $\mathcal{S}_u$, and let $P(z \mid \mathbf{s})$ denote the posterior probability of $z$ given $\mathbf{s} \in \mathcal{S}$. An analyst adopts a label query scheme, for every $n \in [N_q]$, to query the label $Z_n$ of $\mathbf{S}_n \in \mathcal{S}_u$ such that $\mathbf{S}_n$ admits a probability density function (PDF) $p_n(\mathbf{s})$. The label query scheme is a consistent bimodal query if $\lim_{n \to \infty} p_n(\mathbf{s}) = p^*(\mathbf{s})$ where

$$p^*(\mathbf{s}) = 0, \forall \mathbf{s} \in \mathcal{S} \setminus \left(\mathcal{S}_{q_0} \bigcup \mathcal{S}_{q_1}\right), \text{ and } p^*(\mathbf{s}) > 0, \forall \mathbf{s} \in \mathcal{S}_{q_0} \bigcup \mathcal{S}_{q_1}, \tag{5}$$

$$\mathcal{S}_{q_0} = \left\{ \mathbf{s}_{q_0} \,\middle|\, P\left(Z = 0 \mid \mathbf{s}_{q_0}\right) = \max_{\mathbf{s} \in \mathcal{S}} P\left(Z = 0 \mid \mathbf{s}\right) \right\}, \tag{6}$$

$$\mathcal{S}_{q_1} = \left\{ \mathbf{s}_{q_1} \,\middle|\, P\left(Z = 1 \mid \mathbf{s}_{q_1}\right) = \max_{\mathbf{s} \in \mathcal{S}} P\left(Z = 1 \mid \mathbf{s}\right) \right\}. \tag{7}$$

*Remark* 5.3. Def 5.2 considers a label query scheme that only queries the labels of $\mathbf{s}$ with the highest $p\left(Z = 0 \mid \mathbf{s}\right)$ and $p\left(Z = 1 \mid \mathbf{s}\right)$ when $n$ goes to infinity. As $p\left(z \mid \mathbf{s}\right)$ is not directly available, to construct the *consistent bimodal query*, one can use nonparametric regressors to construct a class-probability predictor $Q\left(z \mid \mathbf{s}\right)$ as nonparametric estimates of $P\left(z \mid \mathbf{s}\right), \forall \mathbf{s} \in \mathcal{S}$ and implements the bimodal query to label $\mathbf{s}$ with highest $Q(Z = 0 \mid \mathbf{s})$ or highest $Q(Z = 1 \mid \mathbf{s})$ after $Q(z \mid \mathbf{s})$ converges to $P(z \mid \mathbf{s})$. The authors of (Györfi et al., 2002) prove that when $Q\left(z \mid \mathbf{s}\right)$ is a kernel, KNN or partition estimates with proper smoothing parameters (e.g., bandwidth for the kernel) and labels are sufficiently revealed in the proximity of $\mathbf{s}, \forall \mathbf{s} \in \mathcal{S}$, then $Q\left(z \mid \mathbf{s}\right)$ converges to $P\left(z \mid \mathbf{s}\right)$.

To this end, we introduce the asymptotic property of our framework. We consider normalizing the test statistic in equation 3 as follows,

$$\overline{W}_n = \frac{1}{n} \sum_{i=1}^{n} \log \frac{\hat{P}(Z_i)}{Q_i\left(Z_i \mid \mathbf{S}_i\right)}, (\mathbf{S}_i, Z_i) \sim p_i\left(\mathbf{s}, z\right) = p(z \mid \mathbf{s}) \, p_i\left(\mathbf{s}\right) \tag{8}$$

where $(\mathbf{S}_i, Z_i)$ denotes a feature-label pair returned by a label query scheme when querying the $i$-th label. Next, we state the following theorem.

**Theorem 5.4.** *Let $\mathcal{S}$ be the support of $p\left(\mathbf{s}\right)$ that sample features are collected from and added to an unlabeled set $\mathcal{S}_u$, and let $P\left(z \mid \mathbf{s}\right)$ denote the posterior probability of $z$ given $\mathbf{s} \in \mathcal{S}$. There exists a consistent bimodal query scheme; when an analyst uses such a scheme in the proposed active sequential framework, then, under $H_1$, $\overline{W}_n$ converges to the negation of mutual information (MI), and the converged negated MI lower-bounds the negated MI generated by any $p\left(\mathbf{s}\right)$ subject to $P\left(z \mid \mathbf{s}\right), \forall \mathbf{s} \in \mathcal{S}$. Precisely, there exists a consistent bimodal query leading to the following*

$$\lim_{n \to \infty} \overline{W}_n = -\left(H^*(Z) - H^*\left(Z \mid \mathbf{S}\right)\right) = -I^*\left(\mathbf{S}; Z\right) \leq -I\left(\mathbf{S}; Z\right). \tag{9}$$

$I^*\left(\mathbf{S}; Z\right)$ *is the MI constructed with $(\mathbf{S}, Z) \sim p^*\left(\mathbf{s}, z\right) = P\left(z \mid \mathbf{s}\right) p^*\left(\mathbf{s}\right)$ (See equation 5 for $p^*\left(\mathbf{s}\right)$); $I\left(\mathbf{S}; Z\right)$ is MI constructed with $(\mathbf{S}, Z) \sim p\left(\mathbf{s}, z\right) = P\left(z \mid \mathbf{s}\right) p\left(\mathbf{s}\right)$.*

Recalling the null $H_0$ is rejected when the test statistic $w_n$ in equation 3 is smaller than $\alpha$; hence, the proposed framework, when used with a consistent bimodal query to asymptotically minimize the normalized $w_n$ in equation 3, favorably increases the testing power when $|\mathcal{S}_u|$ is large and $Q(z \mid \mathbf{s})$ is close to $P(z \mid \mathbf{s})$. In Section 5.4, we will analyze the finite-sample performance of the proposed framework considering the approximation error of $Q(z \mid \mathbf{s})$. Additionally, by characterizing the difficulty of a two-sample testing problem with MI, Theorem 5.4 alludes that the proposed framework asymptotically turns the original hard two-sample testing problem with low dependency between $\mathbf{S}$ and $Z$ (low MI), to a simple one by increasing the dependency between $\mathbf{S}$ and $Z$ (high MI).

*Remark* 5.5. Our testing framework is also ***consistent*** under $H_1$ and the same conditions of Theorem 5.4 as $\lim_{n\to\infty} P_1 \left( \prod_{i=1}^n \frac{\hat{P}(Z_i)}{Q_i(Z_i|\mathbf{S}_i)} \leq \alpha \right) = \lim_{n\to\infty} P \left( \overline{W}_n \leq \frac{1}{n} \log(\alpha) \right) = P_1(-I^*(\mathbf{S}, Z) \leq 0) = 1$. The last equality holds due to $I^*(\mathbf{S}, Z) > 0$ under $H_1$.

### 5.4 Finite-sample analysis for the proposed framework

This section analyzes the testing power of the proposed framework in the finite-sample case. Section 5.4.1 and Section 5.4.2 offer metrics that assess the approximation error of $Q(z \mid \mathbf{s})$ and an irreducible Type II error. These metrics together determine the finite-sample testing power. Furthermore, Section 5.4.3 presents an illustrative example of using our framework. In Section 5.4.4, we conduct a finite-sample analysis for the example, incorporating both the metrics that characterize the approximation error and the irreducible Type II error.

#### 5.4.1 Characterizing the approximation error of $Q(z \mid \mathbf{s})$

As our framework constructs the test statistic in equation 2 with the approximation $Q(z \mid \mathbf{s})$, there arises a need to establish a metric for assessing the approximation error of $Q(z \mid \mathbf{s})$ for our finite-sample analysis. To this end, we introduce $\mathrm{KL}^2$-divergence,

**Definition 5.6. ($\mathbf{KL^2}$-divergence)** Let $p_0$ and $q_0$ be two probability density functions on the same support $\mathcal{X}$. Let $f(t) = \log^2(t)$. Then, the $\mathrm{KL}^2$-divergence between $p_0$ and $q_0$ is

$$D_{\mathrm{KL}^2}(q_0 \| p_0) = \mathbb{E}_{\mathbf{X} \sim p_0(\mathbf{x})} \left[ f \left( \frac{q_0(\mathbf{X})}{p_0(\mathbf{X})} \right) \right] = \mathbb{E}_{\mathbf{X} \sim p_0(\mathbf{x})} \left[ \log^2 \left( \frac{q_0(\mathbf{X})}{p_0(\mathbf{X})} \right) \right]. \tag{10}$$

$D_{\mathrm{KL}^2}(q_0 \| p_0)$ is the second moment of the log-likelihood ratio and has been used (see, e.g., (3.1.14) in (Koga et al., 2002)) to understand the behavior of the distribution of $\log \left( \frac{q_0(\mathbf{x})}{p_0(\mathbf{x})} \right)$. We use $D_{\mathrm{KL}^2}(q_0 \| p_0)$ to evaluate the distance between $p(\mathbf{s}, z) = P(z \mid \mathbf{s}) p(\mathbf{s})$ and $q(\mathbf{s}, z) = Q(z \mid \mathbf{s}) p(\mathbf{s})$, which yields the following

$$D_{\mathrm{KL}^2}(q(\mathbf{s}, z) \| p(\mathbf{s}, z)) = \mathbb{E}_{(\mathbf{S}, Z) \sim p(\mathbf{s}, z)} \left[ \log^2 \left( \frac{q(\mathbf{S}, Z)}{p(\mathbf{S}, Z)} \right) \right] = \mathbb{E}_{(\mathbf{S}, Z) \sim p(\mathbf{s}, z)} \left[ \log^2 \left( \frac{Q(Z \mid \mathbf{S})}{P(Z \mid \mathbf{S})} \right) \right]. \tag{11}$$

Remarkably, $D_{\mathrm{KL}^2}(q(\mathbf{s}, z) \| p(\mathbf{s}, z))$ in equation 11 also characterizes the discrepancy between $P(z \mid \mathbf{s})$ and $Q(z \mid \mathbf{s})$ by averaging their log square distance over $\mathcal{S}$; in our main result, we will see that the testing power of the proposed framework depends on $D_{\mathrm{KL}^2}(q(\mathbf{s}, z) \| p(\mathbf{s}, z))$. Additionally, $D_{\mathrm{KL}^2}(q(\mathbf{s}, z) \| p(\mathbf{s}, z))$ is closely related to the typical KL divergence $D_{\mathrm{KL}}(P(z \mid \mathbf{s}) \| Q(z \mid \mathbf{s})) = \mathbb{E}_{(\mathbf{S}, Z) \sim p(\mathbf{s}, z)} \left[ \log \frac{P(Z|\mathbf{S})}{Q(Z|\mathbf{S})} \right]$. This can be seen by expanding equation 11 using the formula $\mathrm{Var}[X] = \mathbb{E}[X^2] - \mathbb{E}^2[X]$ resulting in,

$$D_{\mathrm{KL}^2}(q(\mathbf{s}, z) \| p(\mathbf{s}, z)) = \mathrm{Var}_{(\mathbf{S}, Z) \sim p(\mathbf{s}, z)} \left[ \log \left( \frac{P(Z \mid \mathbf{S})}{Q(Z \mid \mathbf{S})} \right) \right] + [D_{\mathrm{KL}}(P(z \mid \mathbf{s}) \| Q(z \mid \mathbf{s}))]. \tag{12}$$

equation 12 implies that $D_{\mathrm{KL}^2}(q(\mathbf{s}, z) \| p(\mathbf{s}, z))$ not only measures the expected distance between $P(z \mid \mathbf{s})$ and $Q(z \mid \mathbf{s})$ over $\mathcal{S}$ but also the variance of that distance. Similarly, we write

$$D_{\mathrm{KL}^2}(p(\mathbf{s}, z) \| q(\mathbf{s}, z)) = \mathbb{E}_{(\mathbf{S}, Z) \sim q(\mathbf{s}, z)} \left[ \log^2 \left( \frac{P(Z \mid \mathbf{S})}{Q(Z \mid \mathbf{S})} \right) \right] \tag{13}$$

to measure the discrepancy between $p(\mathbf{s}, z)$ and $q(\mathbf{s}, z)$ but with a reverse direction opposed to $D_{\mathrm{KL}^2}(q(\mathbf{s}, z) \| p(\mathbf{s}, z))$.

$D_{\mathbf{KL^2}}(q(\mathbf{s}, z) \| p(\mathbf{s}, z))$ **and** $D_{\mathbf{KL^2}}(p(\mathbf{s}, z) \| q(\mathbf{s}, z))$ **both characterize the approximation error of $Q(z \mid \mathbf{s})$, and we will also see they jointly determine the testing power of the proposed framework in Section 5.4.4.**

### 5.4.2 Characterizing the factor that leads to the irreducible Type II error in finite-sample case

We also introduce another factor influencing testing power, which persists even in the absence of approximation error, i.e., $Q(z \mid \mathbf{s}) = P(z \mid \mathbf{s})$. To see this, we recall the information spectrum introduced in (Han & Verdú, 1993),

**Definition 5.7.** (**Information spectrum (Han & Verdú, 1993)**) Let $(\mathbf{X}, \mathbf{Y})$ be a pair of random variables over the support $\mathcal{X} \times \mathcal{Y}$. Let $p_{\mathbf{XY}}$ denote the joint distribution of $(\mathbf{X}, \mathbf{Y})$, and let $p_{\mathbf{X}}$ and $p_{\mathbf{Y}}$ denote the marginal distributions of $\mathbf{X}$ and $\mathbf{Y}$. Suppose $\{(\mathbf{X}, \mathbf{Y})\}_{i=1}^{n}$ is a sequence of i.i.d random variables for $(\mathbf{X}, \mathbf{Y}) \sim p_{\mathbf{XY}}(\mathbf{x}, \mathbf{y})$. Then, the information spectrum is the probability distribution of the following random variable,

$$\bar{I}(\mathbf{X}^n; \mathbf{Y}^n) = \frac{1}{n} \sum_{i=1}^{n} \log \frac{p_{\mathbf{XY}}(\mathbf{X}_i, \mathbf{Y}_i)}{p_{\mathbf{X}}(\mathbf{X}_i) p_{\mathbf{Y}}(\mathbf{Y}_i)}, \quad (\mathbf{X}, \mathbf{Y}) \sim p_{\mathbf{XY}}(\mathbf{x}, \mathbf{y}) \tag{14}$$

It is easy to see the expectation of $\bar{I}(\mathbf{X}^n; \mathbf{Y}^n)$ is the mutual information $I(\mathbf{X}; \mathbf{Y})$ for $(\mathbf{X}, \mathbf{Y}) \sim p_{\mathbf{XY}}(\mathbf{x}, \mathbf{y})$. Substituting $(\mathbf{X}, \mathbf{Y}) \sim p_{\mathbf{XY}}(\mathbf{x}, \mathbf{y})$ in equation 14 with the feature-label variable pair $(\mathbf{S}, Z) \sim p(\mathbf{s}, z)$ in our two-sample testing problem recovers the (negated) normalizing test statistic in equation 8 with $P(z)$ and $P(z \mid \mathbf{s})$ inserted, i.e., **in the absence of approximation error**.

(Han, 2000) leverages the dispersion of the information spectrum (the distribution of $\bar{I}(\mathbf{X}^n; \mathbf{Y}^n)$) for $\{(\mathbf{X}, \mathbf{Y})\}_{i=1}^{n}$ to quantify the rate that Type II error goes to zero with increasing $n$. Their underlying rationale is that, for a larger variance of $\bar{I}(\mathbf{X}^n; \mathbf{Y}^n)$, the probability of $\bar{I}(\mathbf{X}^n; \mathbf{Y}^n)$ falling outside the acceptance region for an alternative hypothesis also increases, thereby resulting in a slower convergence rate for the Type II error. In our work, we will make use of the variance of the log-likelihood ratio between $p(\mathbf{s}, z)$ and $p(\mathbf{s})p(z)$

$$\mathrm{Var}_{(\mathbf{S};Z) \sim p(\mathbf{s},z)} \bar{I}(\mathbf{S}, Z) = n \mathrm{Var}_{(\mathbf{S};Z)^n \sim p((\mathbf{s},z)^n)} \bar{I}(\mathbf{S}^n, Z^n) = \mathrm{Var}_{(\mathbf{S},Z) \sim p(\mathbf{s},z)} \left[ -\log \frac{P(Z)}{P(Z \mid \mathbf{S})} \right]. \tag{15}$$

Scaling $\mathrm{Var}_{(\mathbf{S},Z) \sim p(\mathbf{s},z)} \bar{I}(\mathbf{S}; Z)$ down by $n$ is the variance of $\bar{I}(\mathbf{S}^n; \mathbf{Z}^n)$, characterizing the the dispersion of the information spectrum for $\{(\mathbf{S}, \mathbf{Z})\}_{i=1}^{n}$ given $n$. $\mathrm{Var}_{(\mathbf{S},Z) \sim p(\mathbf{s},z)} \bar{I}(\mathbf{S}; Z)$ is also known as the **relative entropy variance** (See e.g., (2.29) in (Tan et al., 2014)). It remains present even in the absence of approximation error (i.e., $Q(z \mid \mathbf{s}) = P(z \mid \mathbf{s})$). **As we will see in Section 5.4.4, the persistent $\mathrm{Var}_{(\mathbf{S},Z) \sim p(\mathbf{s},z)} \bar{I}(\mathbf{S}; Z)$ leads to a non-zero Type II error in the finite-sample case.**

### 5.4.3 An example of using the proposed framework

We first introduce the notation that will be used in the ensuing sections. We write $\mathcal{P} = \{A_1, \cdots, A_m\}$ to denote a partition of the support $\mathcal{S}$ of $p(\mathbf{s})$ from which unlabeled sample features in $\mathcal{S}_u$ are generated; in other words, $\bigcup_{i=1}^{m} A_i = \mathcal{S}$. We compare an example of our proposed framework with the baseline, where features are randomly sampled from $\mathcal{S}_u$ and labeled. We quantitatively analyze the testing power of both cases. Both the example and the baseline are detained as follows:

(***An example of using the proposed framework***) Given a label budget $N_q$, $\alpha$, an unlabeled set $\mathcal{S}_u$, a partition $\mathcal{P} = \{A_1, \cdots, A_m\}$, and class priors $\{P(Z = 0 \mid A_1), \cdots, P(Z = 0 \mid A_m)\}$, an analyst initializes $Q(z \mid \mathbf{s})$ with a set of labeled features randomly sampled from $\mathcal{S}_u$, then, she estimates $I(\mathbf{S}; Z \mid A_i)$ by

$$\hat{I}(\mathbf{S}; Z \mid A_i)$$
$$= H(Z \mid A_i) - \hat{H}(Z \mid \mathbf{S}, A_i)$$
$$= -\sum_{z=0}^{1} P(Z = z \mid A_i) \log P(Z = z \mid A_i) + \frac{\sum_{\mathbf{s} \in A_i \bigcap \mathcal{S}_u} \sum_{z=0}^{1} Q(Z = z \mid \mathbf{s}) \log Q(Z = z \mid \mathbf{s})}{|A_i \bigcap \mathcal{S}_u|}, \quad (16)$$

selects $A^* = \arg\max_{A \in \mathcal{P}} \hat{I}(\mathbf{S}; Z \mid A)$, and sequentially constructs the statistic $w_n = \prod_{i=1}^{n} \frac{P(z_i)}{Q(z_i \mid \mathbf{s}_i)}$ by labelling features randomly sampled from $A^* \bigcap \mathcal{S}_u$. The analyst rejects $H_0$ whenever $w_n \leq \alpha$ or retains $H_0$ if the label budget runs out.

(***Baseline test***) Given a label budget $N_q$, $\alpha$, an unlabeled set $\mathcal{S}_u$ and the class prior $P(Z = 0)$, an analyst initializes $Q(z \mid \mathbf{s})$ with a set of labeled features randomly sampled from $\mathcal{S}_u$, then, she sequentially constructs the statistic $w_n = \prod_{i=1}^{n} \frac{P(z_i)}{Q(z_i \mid \mathbf{s}_i)}$ by labelling features randomly sampled from $\mathcal{S}_u$. The analyst rejects $H_0$ whenever $w_n \leq \alpha$ or retains $H_0$ if the label budget runs out.

In the example of using the proposed framework, the class priors $\{P(z \mid A_i)\}$ are given to simplify our analytical results; however, one can estimate these priors with labels in each $A_i$ and use the prior estimates to replace $\{P(z \mid A_i)\}$, and that will not change the main argument of our theorem. In addition, the analyst chooses the partition $A^*$ predicted by $Q(z \mid \mathbf{s})$ to have the highest dependency between $\mathbf{S}$ and $Z$ and only conducts sequential testing with the labeled points in $A^*$. In contrast, the baseline conducts the sequential test entirely the same, except that the analyst queries the labels of features that are randomly generated from $\mathcal{S}_u$. Both the proposed framework and the baseline assert the use of a stable $Q(z \mid \mathbf{s})$ with no updates in the sequential testing; that is sufficient for our analysis as we will see the testing power for the above cases depend on $D_{\text{KL}^2}(q(\mathbf{s}, z) || p(\mathbf{s}, z))$ in equation 11, $D_{\text{KL}^2}(p(\mathbf{s}, z) || q(\mathbf{s}, z))$ in equation 12 and $\text{Var}_{(\mathbf{S}, Z) \sim p(\mathbf{s}, z)} \bar{I}(\mathbf{S}, Z)$ in equation 15

### 5.4.4 Finite-sample analysis for the example

We use $\epsilon_1 = \max_{A \in \mathcal{P}} D_{\text{KL}^2}(q(\mathbf{s}, z) \| p(\mathbf{s}, z) \mid A)$ and $\epsilon_2 = \max_{A \in \mathcal{P}} D_{\text{KL}^2}(p(\mathbf{s}, z) \| q(\mathbf{s}, z) \mid A)$ to capture the maximum approximation error of $Q(z \mid \mathbf{s})$ over the partition $\mathcal{P} = \{A_1, \cdots, A_m\}$, and use $\sigma^2 = \max\{\max_{A \in \mathcal{P}} \text{Var}_{(\mathbf{S}, Z) \sim p(\mathbf{s}, z \mid A)} \bar{I}(\mathbf{S}; Z), \text{Var}_{(\mathbf{S}, Z) \sim p(\mathbf{s}, z)} \bar{I}(\mathbf{S}; Z)\}$ to capture the maximum irreducible Type II error over the same partition $\mathcal{P}$.

We will need to make the following assumptions before presenting our results.

**Assumption 5.8.** (**Maximum mutual information gain**) $\max_{A \in \mathcal{P}} I(\mathbf{S}; Z \mid A) - I(\mathbf{S}; Z) = \Delta \geq 0$.

Assumption 5.8 characterizes the largest MI gain of the proposed framework in the example over the baseline; that is the direct reason for the increased testing power of the proposed framework.

**Assumption 5.9.** (**Sufficient number of unlabeled samples**) $\frac{\sum_{\mathbf{s} \in A \cap \mathcal{S}_u} \mathbb{E}_{Z \sim Q(z \mid \mathbf{s})} \left[\log\left(\frac{Q(Z \mid \mathbf{s})}{P(Z \mid \mathbf{s})}\right)\right]}{|A \cap \mathcal{S}_u|} \approx D_{\text{KL}}(Q(z \mid \mathbf{s}) \| P(z \mid \mathbf{s}) \mid A), \forall A \in \mathcal{P}$.

Even though we typically have access to only a finite number of unlabeled samples in real-world scenarios, this number is usually quite large and affordable for many applications. Hence, similar to (Hanneke & Yang, 2015), Assumption 5.9 assumes a sufficient supply of unlabeled samples to simplify the analysis and concentrate solely on the number of labels needed for the proposed framework in the example.

Now, we present our theorem to address the testing power of the framework in the example and the baseline test in the finite-sample case.

**Theorem 5.10.** *Under Assumption 5.85.9, the proposed framework in the example with a label budget $N_q$ and $\alpha$ has a testing power of approximately at least*

$$\Phi\left(\frac{\frac{\log\alpha}{\sqrt{N_q}} + \sqrt{N_q}\left(I\left(\mathbf{S};Z\right) + \Delta - 2\sqrt{\epsilon_1} - \sqrt{\epsilon_2}\right)}{\left(\epsilon_1 + \sigma^2 + 2\sigma\sqrt{\epsilon_1}\right)^{1/2}}\right); \tag{17}$$

*and the baseline test with features randomly sampled from $\mathcal{S}_u$ and labeled has a testing power of approximately at least*

$$\Phi\left(\frac{\frac{\log\alpha}{\sqrt{N_q}} + \sqrt{N_q}\left(I\left(\mathbf{S};Z\right) - \sqrt{\epsilon_1}\right)}{\left(\epsilon_1 + \sigma^2 + 2\sigma\sqrt{\epsilon_1}\right)^{1/2}}\right). \tag{18}$$

equation 17 and equation 18 state approximate testing power's lower bounds for the proposed framework in the example and the baseline test. We can observe that

- Given $\alpha$, then, a large budget $N_q$, and small approximation errors characterized by $\epsilon_1$, increase the two testing power's lower-bounds of the proposed framework and the baseline, as structured similarly in equation 17 and equation 18.

- Comparing equation 17 for the proposed framework to the equation 18 for the baseline, the extra $\Delta$ is ascribed to the maximum power gain, and $\sqrt{\epsilon_1} + \sqrt{\epsilon_2}$ accounts for the diminishing of the maximum power gain in selecting a $A^* \in \mathcal{P}$ that does not have the highest MI over $A \in \mathcal{P}$.

- When the approximation errors $\epsilon_1 = 0$ and/or $\epsilon_2 = 0$, both testing power's lower-bounds are decreased by a factor of $\sigma$, resulting in the irreducible Type II error.

- When the maximum MI gain $\Delta$ can compensate the approximation error of $Q\left(z \mid \mathbf{s}\right)$ being larger than $\sqrt{\epsilon_1} + \sqrt{\epsilon_2}$, our framework in the example has higher testing power's lower bound than the baseline test given the same label budget $N_q$ and $\alpha$.

## 6 Experimental Results

We have proposed a practical instantiation of the framework, and its algorithmic description BQ-AST is presented in Algorithm 1. In this section, we compare the BQ-AST with a sequential testing baseline (Lhéritier & Cazals, 2018) that uses the same statistic in equation 2, but the baseline labels features randomly sampled from the unlabeled set $\mathcal{S}_u$. In addition, we build $Q\left(z \mid \mathbf{s}\right)$ for the test statistic in equation 2 using logistic regression, SVM, or KNN classifiers; we set $N_0 = 10$ for the number of label queries used to initialize $Q\left(z \mid \mathbf{s}\right)$, and set significance level $\alpha = 0.05$.

### 6.1 Experiments on Synthetic Datasets

Our first suite of experiment results is generated from synthetic data. We create synthetic datasets that comprise two samples of data to simulate cases under the null hypothesis $H_0$ and the alternative hypothesis $H_1$; the data for the first sample $(Z = 0)$ is generated from $p\left(\mathbf{s} \mid Z = 0\right) \equiv \mathcal{N}\left((-\delta, 0), I\right)$ and the data for the second sample $(Z = 1)$ is generated from $p\left(\mathbf{s} \mid Z = 1\right) \equiv \mathcal{N}\left((\delta, 0), I\right)$. In addition, we set $P(Z = 0)$ from 0.5 to 0.8 to vary the ratio of the data sizes for two samples. For the simulations of the data under $H_0$, we set $\delta = 0$, implying there is no difference between the distributions that generate the two samples; for the simulations of the data under $H_1$, we vary $\delta$ from 0.2 to 0.5 to simulate two samples from small to high discrepancy under $H_1$. Having constructed the data-generating process, we simulate 200 cases of data for each pair of $P(Z = 0)$ and $\delta$ under $H_1$, and simulate 500 cases of data for each pair of $P(Z = 0)$ and $\delta = 0$ under $H_0$. Each case of data is of size 2000 with labels masked, resulting in an unlabeled set $\mathcal{S}_u$ with $|\mathcal{S}_u| = 2000$. The proposed test actively and sequentially labels feature in $\mathcal{S}_u$ to test the difference between the two samples.

Figure 2 presents the empirical Type I errors: when $H_0$ is true, the probability of the proposed test mistakenly predicting the two samples is generated under $H_1$. As observed, the empirical Type I errors are all smaller than $\alpha = 0.05$ for using various classifiers and label budgets in the experiments; this provides empirical evidence for Theorem 5.1, which states that the Type I error is controlled to be smaller than the significance level $\alpha$.

Table-1 presents the empirical Type II errors: when $H_1$ is true, the probability of the proposed test and the baseline test mistakenly predicting the two samples are generated under $H_0$. Table 2 presents the average label queried spent to reject $H_0$ when $H_1$ is true. We can observe from Table-1 that the proposed test produces lower Type II errors than that of the baseline under different classifiers and label budgets; furthermore, in Table 2, we observe the proposed test spends a smaller number of label queries than the baseline test. Additionally, we run a two-sample t-test to assess the mean difference of label query numbers generated by 200 runs using both methods. The resultant $p$-values, truncated to the last 6 decimal places, all equate to zero, indicating that the label spent by our framework is

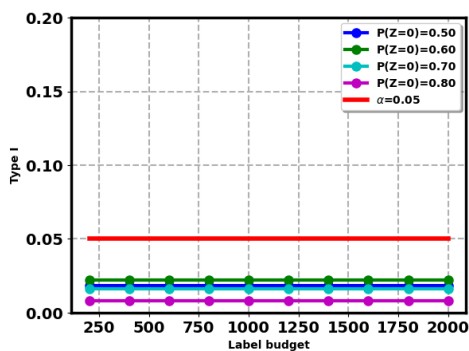

Figure 2: Under $H_0$ in which $\delta = 0$, empirical Type I errors of the proposed test for different $P(Z = 0)$ when using the logistic regression to build $Q(z \mid \mathbf{s})$. All Type I errors are smaller than $\alpha = 0.05$, which agrees with Theorem 5.1.

statistically smaller than the baseline test. All these observations demonstrate that, under $H_1$, the proposed test labels the features that have a high dependency on labels to effectively decrease the Type II error and reduce the number of label queries needed to reject $H_0$.

Table 1: Under $H_1$, **Type II errors** of conducting the proposed/baseline with various classifiers and label budgets for the synthetic data generated by setting $\delta = 0.2$ and different class priors $P(Z = 0)$. Due to the active query, our test produces lower Type II errors than the baseline for various label budgets.

| $P(Z=0)$ | | Logistic | | | | | KNN | | | | |
|---|---|---|---|---|---|---|---|---|---|---|---|
| | Label budget | 200 | 400 | 600 | 800 | 1000 | 200 | 400 | 600 | 800 | 1000 |
| 0.5 | Baseline | 0.82 | 0.53 | 0.29 | 0.11 | 0.04 | 0.95 | 0.77 | 0.50 | 0.28 | 0.14 |
| | Proposed | **0.16** | **0.02** | **0.00** | **0.00** | **0.00** | **0.49** | **0.17** | **0.06** | **0.03** | **0.01** |
| 0.6 | Baseline | 0.80 | 0.50 | 0.23 | 0.12 | 0.06 | 0.95 | 0.77 | 0.48 | 0.29 | 0.14 |
| | Proposed | **0.26** | **0.06** | **0.01** | **0.01** | **0.01** | **0.59** | **0.26** | **0.09** | **0.03** | **0.01** |
| 0.7 | Baseline | 0.81 | 0.56 | 0.34 | 0.22 | 0.10 | 0.96 | 0.81 | 0.58 | 0.36 | 0.28 |
| | Proposed | **0.26** | **0.04** | **0.01** | **0.01** | **0.01** | **0.71** | **0.33** | **0.14** | **0.04** | **0.02** |
| 0.8 | Baseline | 0.88 | 0.73 | 0.56 | 0.35 | 0.21 | 0.98 | 0.90 | 0.77 | 0.59 | 0.48 |
| | Proposed | **0.38** | **0.10** | **0.04** | **0.03** | **0.02** | **0.80** | **0.50** | **0.28** | **0.16** | **0.10** |

Table 2: Under $H_1$, **average number of label queries** needed to reject $H_0$ for the proposed/baseline test using various classifiers and label budgets in the synthetic data generated by setting $\delta = 0.2$ and different class priors $P(Z = 0)$. Due to the active query, our test spends fewer label queries to reject $H_0$ than the baseline for various label budgets.

| $P(Z=0)$ | | Logistic | | | | | KNN | | | | |
|---|---|---|---|---|---|---|---|---|---|---|---|
| | Label budget | 200 | 400 | 600 | 800 | 1000 | 200 | 400 | 600 | 800 | 1000 |
| 0.5 | Baseline | 183.5±41 | 319.7±113 | 399.1±183 | 438.1±233 | 451.4±257 | 198.1±10 | 374.4±61 | 500.4±132 | 578.1±201 | 619.5±254 |
| | Proposed | **95.3±64** | **108.1±92** | **108.6±93** | **108.6±93** | **108.6±93** | **162.1±50** | **223.1±116** | **240.8±149** | **249.7±173** | **252.8±184** |
| 0.6 | Baseline | 182.3±41 | 312.4±116 | 386.0±184 | 419.7±231 | 439.0±266 | 196.7±16 | 373.7±66 | 499.3±134 | 578.9±206 | 619.7±256 |
| | Proposed | **107.9±70** | **134.2±114** | **142.3±136** | **143.7±142** | **144.7±147** | **166.3±48** | **246.8±123** | **282.2±175** | **294.3±200** | **296.6±207** |
| 0.7 | Baseline | 184.0±41 | 323.3±113 | 415.5±188 | 472.2±252 | 505.0±299 | 198.3±11 | 378.5±58 | 520.0±127 | 613.4±199 | 678.1±268 |
| | Proposed | **120.4±67** | **143.4±104** | **147.6±117** | **149.0±122** | **150.0±128** | **178.0±43** | **282.2±117** | **327.4±173** | **345.9±207** | **351.7±222** |
| 0.8 | Baseline | 190.8±31 | 351.7±96 | 479.6±172 | 571.1±245 | 628.0±306 | 199.0±8 | 386.6±47 | 555.0±106 | 689.5±175 | 798.4±253 |
| | Proposed | **134.7±64** | **174.8±118** | **189.5±151** | **195.6±170** | **199.7±186** | **184.4±36** | **310.2±111** | **387.7±186** | **434.7±247** | **462.6±293** |

We present the average number of label queries spent for two samples with small to big discrepancies under $H_1$ in Table 3. A small discrepancy between two samples indicates a more difficult two-sample testing problem than one with a large discrepancy between the two samples, as a two-sample test requires more data to test the existence of the small discrepancy. Table 3 shows that the proposed active sequential test spends fewer labels to reject $H_0$ when increasing the mean discrepancy $\delta$ between two samples, which demonstrates the proposed sequential test automatically adapts the number of label queries to the problem's complexity.

Table 3: Under $H_1$ and label budget $N_q = 1000$, the **average number of label queries** needed to reject $H_0$ for different $\delta$. When the mean difference $\delta$ increases between two samples, both our active sequential test and the baseline test reject $H_0$ with a reduced number of label queries spent, exhibiting the sequential test's benefit that the tests adapt the label queries to the problem's complexity. Due to the active query, our test spends fewer label queries to reject $H_0$ than the baseline for various $\delta$.

| $P(Z=0)$ | | Logistic | | | | KNN | | | |
|---|---|---|---|---|---|---|---|---|---|
| | $\delta$ | 0.2 | 0.3 | 0.4 | 0.5 | 0.2 | 0.3 | 0.4 | 0.5 |
| 0.5 | Baseline | 451.4±257 | 178.3±105 | 101.0±58 | 63.9±32 | 619.5±254 | 287.8±129 | 167.4±70 | 116.8±43 |
| | Proposed | **108.6±93** | **37.3±22** | **24.3±10** | **19.7±5** | **252.8±184** | **109.5±64** | **72.2±33** | **54.9±20** |
| 0.6 | Baseline | 439.0±266 | 175.3±118 | 96.9±65 | 65.5±40 | 619.7±256 | 289.8±130 | 170.2±72 | 116.2±47 |
| | Proposed | **144.7±147** | **40.5±30** | **24.9±11** | **20.1±7** | **296.6±207** | **134.3±88** | **84.3±43** | **58.3±25** |
| 0.7 | Baseline | 505.0±299 | 223.6±145 | 115.7±70 | 75.7±47 | 678.1±268 | 349.3±178 | 198.2±93 | 133.3±56 |
| | Proposed | **150.0±128** | **57.1±42** | **32.3±21** | **22.2±8** | **351.7±222** | **160.2±107** | **94.0±54** | **67.0±30** |
| 0.8 | Baseline | 628.0±306 | 278.1±177 | 149.3±95 | 94.8±56 | 798.4±253 | 470.3±223 | 268.7±126 | 176.3±81 |
| | Proposed | **199.7±186** | **66.7±41** | **40.0±22** | **29.4±15** | **462.6±293** | **198.8±143** | **115.7±65** | **83.8±46** |

## 6.2 Experiments on MNIST

In addition to the synthetic datasets, We simulate the cases of $H_0$ and $H_1$ with MNIST (LeCun, 1998). To create a case for $H_0$, we randomly pick one digit category from 0-9, then randomly sample images from the selected digit category, and lastly divide the images to sample zero ($Z = 0$) and one ($Z = 1$) based on a pre-defined class prior $P(Z = 0)$; for each case, the two samples contain data from the same digit, but the digit categories could be different over cases. To create a case for $H_1$, we randomly pick two different digit categories from 0-9, then sample images from one digit category and place the images to sample zero ($Z = 0$); to create sample one ($Z = 1$), we sample images from the two digits, mix the sampled images, and place them to sample one. We set the mixture ratio 0.7, meaning there are roughly 30% data in sample one generated from a distribution different from sample zero. We also adjust $P(Z = 0)$ to create cases with different ratios for the size of sample zero over sample one for $H_1$. We produce 500 cases for $H_0$ and 200 cases for $H_1$ with the stated procedure for each $P(Z = 0)$ that ranges from 0.5 to 0.8; each case comprises an unlabeled set $\mathcal{S}_u$ with a size of 2000 and its corresponding

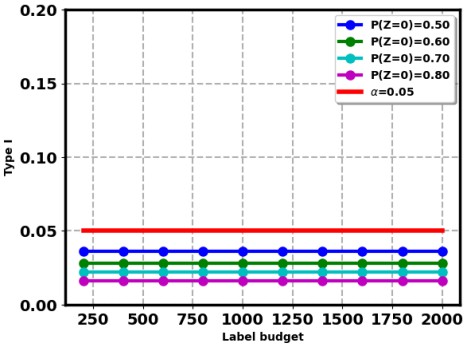

Figure 3: Empirical Type I errors of the proposed test for different $P(Z = 0)$ in the MNIST experiment. SVM is used to build $Q(z \mid \mathbf{s})$. All Type I errors are smaller than $\alpha = 0.05$, which agrees with Theorem 5.1.

labels that are unknown to an analyst. Instead of using the raw data in the created cases, we projected the MNIST data to a 28-dimensional space by a convolutional autoencoder before conducting the two-sample testing.

We first present the empirical Type I errors in Figure 3. We use the support vector machine (SVM) to build $Q(z \mid \mathbf{s})$ to generate the results. As observed, all the Type I errors are smaller than $\alpha = 0.05$, which agrees with Theorem 5.1. In addition, we present the Type II errors, as shown in Table 4. The proposed test generates smaller Type II errors than the baseline sequential test for various classifiers, label budgets, and $P(Z = 0)$, implying the proposed sequential testing combined with the active query is effective. This is further corroborated by Table 5 that exhibits the average number of label queries needed to reject $H_0$;

the proposed test spent fewer label queries than the baseline test to reject $H_0$. We additionally run a two-sample t-test to statistically compare the mean difference between the label query numbers generated by both methods. The resultant $p$-values, truncated to the last 6 decimal places, all equate to zero, indicating that the label spent by our framework is statistically smaller than the baseline test in the MNIST experiment.

Table 4: Under $H_1$, **Type II errors** of conducting the proposed/baseline with various classifiers and label budgets for MNIST and different class priors $P(Z = 0)$. Due to the active query, our test produces lower Type II errors than the baseline for various label budgets.

| $P(Z=0)$ | | Logistic | | | | | SVM | | | | | KNN | | | | |
|---|---|---|---|---|---|---|---|---|---|---|---|---|---|---|---|---|
| | Label budget | 200 | 400 | 600 | 800 | 1000 | 200 | 400 | 600 | 800 | 1000 | 200 | 400 | 600 | 800 | 1000 |
| 0.5 | Baseline | 0.65 | 0.21 | 0.02 | 0.01 | 0.01 | 0.59 | 0.07 | **0.00** | **0.00** | **0.00** | 0.84 | 0.43 | 0.15 | 0.07 | 0.03 |
| | Proposed | **0.12** | **0.01** | 0.01 | **0.00** | **0.00** | **0.12** | **0.03** | 0.01 | 0.01 | **0.00** | **0.10** | **0.01** | **0.01** | **0.00** | **0.00** |
| 0.6 | Baseline | 0.59 | 0.16 | 0.02 | 0.01 | 0.01 | 0.55 | 0.04 | **0.00** | **0.00** | **0.00** | 0.89 | 0.43 | 0.15 | 0.06 | 0.03 |
| | Proposed | **0.01** | **0.00** | **0.00** | **0.00** | **0.00** | **0.06** | **0.01** | **0.00** | **0.00** | **0.00** | **0.06** | **0.02** | **0.01** | **0.01** | **0.00** |
| 0.7 | Baseline | 0.58 | 0.21 | 0.04 | 0.01 | **0.00** | 0.67 | 0.15 | 0.01 | **0.00** | **0.00** | 0.91 | 0.58 | 0.29 | 0.10 | 0.04 |
| | Proposed | **0.00** | **0.00** | **0.00** | **0.00** | **0.00** | **0.10** | **0.01** | **0.00** | **0.00** | **0.00** | **0.12** | **0.03** | **0.01** | **0.00** | **0.00** |
| 0.8 | Baseline | 0.66 | 0.24 | 0.04 | 0.01 | 0.01 | 0.77 | 0.32 | 0.10 | 0.01 | 0.01 | 0.95 | 0.71 | 0.47 | 0.27 | 0.12 |
| | Proposed | **0.00** | **0.00** | **0.00** | **0.00** | **0.00** | **0.06** | **0.01** | **0.00** | **0.00** | **0.00** | **0.14** | **0.03** | **0.01** | **0.01** | **0.00** |

Table 5: Under $H_1$, **average number of label queries** needed to reject $H_0$ for the proposed/baseline test using various classifiers and label budgets for MNIST and different class priors $P(Z = 0)$. Due to the active query, our test spends fewer label queries to reject $H_0$ than the baseline for various label budgets.

| $P(Z=0)$ | | Logistic | | | | | SVM | | | | | KNN | | | | |
|---|---|---|---|---|---|---|---|---|---|---|---|---|---|---|---|---|
| | Label budget | 200 | 400 | 600 | 800 | 1000 | 200 | 400 | 600 | 800 | 1000 | 200 | 400 | 600 | 800 | 1000 |
| 0.5 | Baseline | 165.3±56 | 251.7±126 | 267.9±150 | 270.7±158 | 271.7±162 | 175.4±39 | 229.9±93 | 233.0±99 | 233.0±99 | 233.0±99 | 187.0±30 | 311.3±93 | 359.1±141 | 376.8±167 | 384.6±185 |
| | Proposed | **90.4±62** | **99.8±84** | **101.0±89** | **101.5±92** | **101.5±92** | **93.5±55** | **106.5±87** | **109.4±98** | **110.5±105** | **110.7±106** | **89.4±51** | **97.9±75** | **99.9±84** | **100.1±86** | **100.1±86** |
| 0.6 | Baseline | 160.8±59 | 233.2±125 | 247.5±148 | 249.3±154 | 250.3±158 | 173.5±39 | 226.8±95 | 229.8±101 | 229.8±101 | 229.8±101 | 187.5±31 | 315.1±93 | 363.9±142 | 379.8±166 | 385.4±178 |
| | Proposed | **61.7±43** | **61.7±43** | **61.7±43** | **61.7±43** | **61.7±43** | **79.4±48** | **83.2±60** | **83.4±62** | **83.4±62** | **83.4±62** | **85.0±49** | **90.9±68** | **94.0±85** | **95.6±95** | **96.6±103** |
| 0.7 | Baseline | 160.3±59 | 234.8±128 | 255.2±161 | 257.8±167 | 258.0±168 | 174.6±45 | 252.1±109 | 264.6±130 | 265.4±133 | 265.4±133 | 188.4±31 | 330.7±94 | 415.7±162 | 451.8±206 | 463.2±225 |
| | Proposed | **46.2±28** | **46.2±28** | **46.2±28** | **46.2±28** | **46.2±28** | **74.7±56** | **82.0±76** | **83.0±81** | **83.0±81** | **83.0±81** | **89.3±54** | **101.5±85** | **104.6±98** | **105.1±101** | **105.1±101** |
| 0.8 | Baseline | 163.9±58 | 243.3±126 | 268.1±163 | 273.1±175 | 275.3±183 | 92.6±16 | 148.5±52 | 167.3±76 | 171.7±85 | 172.6±88 | 192.8±25 | 357.3±76 | 471.8±146 | 540.7±210 | 575.4±255 |
| | Proposed | **34.8±17** | **34.8±17** | **34.8±17** | **34.8±17** | **34.8±17** | **77.2±55** | **81.6±68** | **82.3±72** | **82.3±72** | **82.3±72** | **104.8±54** | **116.9±81** | **119.1±90** | **120.1±96** | **120.3±98** |

## 6.3 Experiments on An Alzheimer's Disease Dataset

Table 6: Under $H_1$, **Type II errors** of conducting the proposed/baseline with various classifiers and label budgets for ADNI and different class priors $P(Z = 0)$. Due to the active query, our test produces lower Type II errors than the baseline for various label budgets.

| $P(Z=0)$ | | Logistic | | | | | SVM | | | | | KNN | | | | |
|---|---|---|---|---|---|---|---|---|---|---|---|---|---|---|---|---|
| | Label budget | 100 | 200 | 300 | 400 | 500 | 100 | 200 | 300 | 400 | 500 | 100 | 200 | 300 | 400 | 500 |
| 0.5 | Baseline | 0.32 | 0.06 | 0.01 | **0.00** | **0.00** | 0.67 | 0.17 | 0.02 | **0.00** | **0.00** | 0.72 | 0.49 | 0.25 | 0.13 | 0.04 |
| | Proposed | **0.10** | **0.01** | **0.00** | **0.00** | **0.00** | **0.24** | **0.03** | **0.01** | **0.00** | **0.00** | **0.21** | **0.04** | **0.00** | **0.00** | **0.00** |
| 0.6 | Baseline | 0.35 | 0.04 | **0.00** | **0.00** | **0.00** | 0.62 | 0.15 | 0.01 | **0.00** | **0.00** | 0.73 | 0.25 | 0.06 | 0.01 | **0.00** |
| | Proposed | **0.07** | **0.00** | **0.00** | **0.00** | **0.00** | **0.18** | **0.04** | 0.03 | **0.00** | **0.00** | **0.10** | **0.01** | **0.00** | **0.00** | **0.00** |
| 0.7 | Baseline | 0.40 | 0.10 | 0.01 | **0.00** | **0.00** | 0.65 | 0.21 | 0.06 | **0.00** | **0.00** | 0.81 | 0.36 | 0.12 | 0.04 | 0.02 |
| | Proposed | **0.11** | **0.03** | **0.00** | **0.00** | **0.00** | **0.32** | **0.07** | **0.02** | 0.01 | 0.01 | **0.25** | **0.04** | **0.01** | **0.00** | **0.00** |
| 0.8 | Baseline | 0.52 | 0.23 | 0.07 | 0.01 | **0.00** | 0.89 | 0.53 | 0.27 | 0.07 | 0.02 | 0.90 | 0.59 | 0.28 | 0.16 | 0.07 |
| | Proposed | **0.28** | **0.01** | **0.00** | **0.00** | **0.00** | **0.49** | **0.15** | **0.06** | **0.03** | **0.01** | **0.38** | **0.10** | **0.03** | **0.01** | **0.01** |

We demonstrate the utility of the proposed test in a clinical application using data from the Alzheimer's Disease Neuroimaging Initiative (ADNI) database (Jack Jr et al., 2008). The ADNI study protocol was approved by local institutional review boards (IRB). All the personal information in the data provided to researchers has been removed. The motivation for applying the proposed test to Alzheimer's disease research is as follows. Amyloid has been linked to the development of Alzheimer's disease; identifying the amount of amyloid in the human brain is an important step in predicting the progression of Alzheimer's disease. To measure the amyloid level, an expensive CT scan is required used to assess the amyloid deposition in the brain. A useful replacement would be an easy-to-measure and inexpensive replacement for the amyloid to indicate the progression of Alzheimer's disease. In the following experiments, we considered using digital test results that include five cognition measurement scores of participants as a replacement. To verify if the

digital test results are suitable replacements, clinicians are seeking an approach to test the independence between the digital test results and the amyloid amount with a limited number of expensive CT scans to measure the amyloid levels. We use a binary version of the amyloid level where $Z = 0$ and $Z = 1$ suggest low and high amyloid depositions in the brain respectively; we can now formulate a two-sample test and use the proposed scheme. As the results show, our proposed test is endowed with sequential decision-making and active label query, resulting in fewer CT scans needed compared with the conventional sequential test.

Table 7: Under $H_1$, **average number of label queries** needed to reject $H_0$ for the proposed/baseline test using various classifiers and label budgets for ADNI and different class priors $P(Z = 0)$. Due to the active query, our test spends fewer label queries to reject $H_0$ than the baseline for various label budgets.

| $P(Z=0)$ | | Logistic | | | | | SVM | | | | | KNN | | | | |
|---|---|---|---|---|---|---|---|---|---|---|---|---|---|---|---|---|
| | Label budget | 100 | 200 | 300 | 400 | 500 | 100 | 200 | 300 | 400 | 500 | 100 | 200 | 300 | 400 | 500 |
| 0.5 | Baseline | 68.1±29 | 83.7±52 | 85.5±57 | 85.6±57 | 85.6±57 | 87.0±22 | 127.2±55 | 135.4±69 | 136.1±70 | 136.1±70 | 86.2±26 | 145.8±66 | 181.8±100 | 199.5±124 | 207.3±138 |
| | Proposed | **43.9±29** | **47.1±36** | **47.1±37** | **47.1±37** | **47.1±37** | **64.0±28** | **75.1±47** | **76.5±51** | **76.6±52** | **76.6±52** | **69.3±22** | **76.6±37** | **77.8±41** | **77.8±41** | **77.8±41** |
| 0.6 | Baseline | 68.4±29 | 84.0±51 | 86.1±57 | 86.1±57 | 86.1±57 | 85.0±23 | 121.0±55 | 127.5±67 | 127.5±67 | 127.5±67 | 92.7±15 | 140.3±51 | 153.0±69 | 156.0±77 | 156.3±78 |
| | Proposed | **43.9±29** | **45.3±32** | **45.3±32** | **45.3±32** | **45.3±32** | **61.3±26** | **70.0±44** | **72.9±54** | **74.5±61** | **74.5±61** | **60.8±20** | **64.4±30** | **64.4±30** | **64.4±30** | **64.4±30** |
| 0.7 | Baseline | 72.3±29 | 95.6±58 | 100.9±70 | 101.1±70 | 101.1±70 | 86.5±23 | 126.7±57 | 139.0±76 | 141.3±82 | 141.3±82 | 94.7±13 | 153.5±49 | 176.5±77 | 183.7±90 | 186.1±97 |
| | Proposed | **50.6±29** | **56.6±43** | **57.1±45** | **57.1±45** | **57.1±45** | **68.8±29** | **85.1±53** | **89.0±63** | **90.0±67** | **90.5±69** | **68.6±24** | **78.5±42** | **79.9±47** | **79.9±47** | **79.9±47** |
| 0.8 | Baseline | 78.1±28 | 115.4±65 | 128.5±86 | 132.5±95 | 132.9±96 | 95.9±13 | 166.9±47 | 204.8±81 | 219.6±101 | 222.8±108 | 97.6±8 | 171.0±43 | 215.3±79 | 235.8±106 | 247.6±126 |
| | Proposed | **63.6±32** | **72.0±44** | **72.2±45** | **72.2±45** | **72.2±45** | **80.1±26** | **108.6±57** | **118.1±75** | **121.9±86** | **124.0±93** | **80.7±21** | **98.3±46** | **102.4±57** | **103.9±63** | **104.9±68** |

The obtained ADNI data contains both digital test results and the amyloid amount of participants. We use the cut-off value suggested by ADNI and binarize the amyloid amount to create two-sample cases where **s** denote a vector of cognition measurement scores and $z$ denotes low or high amyloid amount for the participants. We create 200 data cases for each $P(Z = 0)$ that ranges from 0.5 to 0.8; these cases are simulations for $H_1$, and each case comprises an unlabeled set $\mathcal{S}_u$ with a size of 1000 and its corresponding labels that are unknown to an analyst.

Table 6 and Table 7 present the results of empirical Type II errors and the average number of label queries needed to reject $H_0$. Our proposed test has Type II errors decreased by 58% and saves on label queries by 62% at most compared with the baseline test with the same label budgets. Additionally, we run a two-sample t-test to statistically compare the mean difference between the label query numbers generated by both methods. The resultant $p$-values, truncated to the last 6 decimal places, all equate to zero; this indicates that the label savings are statistically significant.

## 7 Conclusion

We propose an *active sequential two-sample testing framework* that sequentially and actively labels the data to increase the testing power and adapt the number of label queries to the problem's complexity. We provide both finite-sample and asymptotic analysis of the proposed framework; the framework's benefit is characterized by the change of the mutual information between feature and label variables over a random labeling scheme in both finite-sample and asymptotic cases. Moreover, we suggest an instantiation of the framework, in which we adopt the bimodal query that labels the features predicted by a classifier to have the highest class one or zero probabilities. Our experiments on synthetic data, MNIST, and an Alzheimer's Disease dataset demonstrate the effectiveness of the suggested instantiation of the proposed framework.

## Acknowledgement

This work was funded in part by Office of Naval Research grant N00014-21-1-2615 and by the National Science Foundation (NSF) under grants CNS-2003111, and CCF-2048223.

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

# A  Proof of Theorem 5.1 and Its Preliminaries

## A.1  Some statistical preliminaries

In probability theory, a sequence $\{X_0, \cdots, X_n\}$ of random variables is called *martingale* if at a particular time, the expectation of the next random variable is equivalent to the present observation; this is formally defined as follows,

**Definition A.1.** (*Martingale*) A sequence of random variables $\{X_0, \cdots, X_n\}$ is a *martingale* if, for any $n \geq 0$,

$$\mathbb{E}\left[|X_n|\right] \leq \infty \tag{19}$$

$$\mathbb{E}\left[X_{n+1}|X_0, \cdots, X_n\right] = X_n \tag{20}$$

We refer interested readers to (Aaditya Ramdas, 2018) for a complete introduction to the martingale and its related properties.

Next, we state Ville's maximal inequalityVille (1939), which will be applied to prove Theorem 5.1.

**Theorem A.2.** *(Ville's Maximal Inequality Ville (1939)): If $\{X_n\}$ is a nonnegative martingale, then for any $c > 0$, we have*

$$P\left(\sup_{n\geq 0} X_n > c\right) \leq \frac{\mathbb{E}\left[X_0\right]}{c} \tag{21}$$

Ville's maximal inequality gives a probability upper bound for the event that the martingale crosses a threshold $c$; it is a sequential extension of Markov's inequality.

## A.2 Proof of Theorem 5.1

*Proof.* Our proof comprises proving the following two ordered parts:

(1) The first part is to demonstrate that, under the null hypothesis $H_0$, the independence between unqueried label random variables and the corresponding feature random variables still holds following the adaptive label query. In particular, Under $H_0$, the feature and label variables $\mathbf{S}_i$ and $Z_i$ used to construct the test statistic in equation 3 in the proposed framework are independent $\forall i \in [N_q]$.

(2) In the second part, we consider $\tilde{W}_n = \prod_{i=1}^n \frac{P(Z_i)}{Q_i(Z_i | \mathbf{S}_i)}$, which is the test statistic in equation 2 with true class prior $P(z)$ plugged in. Moving forward, the second part is to demonstrate the following inequalities under $H_0$

$$P_0\left(\exists n \in [N_q], W_n = \prod_{i=1}^n \frac{\hat{P}(Z_i)}{Q_i(Z_i \mid \mathbf{S}_i)} \leq \alpha\right) \leq P_0\left(\exists n \in [N_q], \tilde{W}_n = \prod_{i=1}^n \frac{P(Z_i)}{Q_i(Z_i \mid \mathbf{S}_i)} \leq \alpha\right) \leq \alpha. \tag{22}$$

equation 22 immediately implies that the Type I error of our proposed framework is upper-bounded by $\alpha$.

- **Proof for the first part**

  We write $\mathcal{S}_u$ and $\mathcal{Z}_u$ to denote the sets of original unlabeled feature variables on an analyst's hand and unrevealed label variables provided by an oracle. We write $\mathcal{S}_i^l$ and $\mathcal{Z}_i^l$ to denote the sets of the labeled feature and the corresponding label variables after including the i-th $(\mathbf{S}_i, Z_i)$ to construct the statistic in equation 3. We use $\mathcal{S}_i^u = \mathcal{S}_u \setminus \mathcal{S}_i^l$ and $\mathcal{Z}_i^u = \mathcal{Z}_u \setminus \mathcal{Z}_i^l$ to denote their complements that comprise unlabeled feature and unrevealed label variables. In particular, we use $\mathcal{S}_0^l$ and $\mathcal{Z}_0^l$ to denote the feature and label variable sets used to initialize $Q_1(z \mid \mathbf{s})$ in the first place; $\mathcal{S}_0^u = \mathcal{S}_u \setminus \mathcal{S}_0^l$ and $\mathcal{Z}_0^u = \mathcal{Z}_u \setminus \mathcal{Z}_0^l$ are their complements that comprise unlabeled feature and unrevealed label variables. $H_0$ being true implies $\mathcal{S}_u \perp\!\!\!\perp \mathcal{Z}_u$. In our setting, an analyst randomly samples features and labels them to build $\mathcal{S}_0^l$ and $\mathcal{Z}_0^l$, implying $\mathcal{S}_0^l \perp\!\!\!\perp \mathcal{Z}_0^l$ and $\mathcal{S}_0^u \perp\!\!\!\perp \mathcal{Z}_0^u$ when $H_0$ is true. In the following, we employ the induction method to prove $\mathbf{S}_i$ and $Z_i$ are independent $\forall i \in [N_q]$.

  **Base case** $(i = 1)$: Under $H_0$, we have $\mathcal{S}_0^l \perp\!\!\!\perp \mathcal{Z}_0^l$ and $\mathcal{S}_0^u \perp\!\!\!\perp \mathcal{Z}_0^u$. The analyst first initializes $Q_1(z \mid \mathbf{s})$ with $\mathcal{S}_0^l \perp\!\!\!\perp \mathcal{Z}_0^l$ before starting the sequential testing. Subsequently, the analyst makes a query on a label based on the prediction of $Q_1(z \mid \mathbf{s})$ and includes the first variable pair $(\mathbf{S}_1, Z_1)$ to construct the test statistic. That immediately implies $\mathbf{S}_1 \perp\!\!\!\perp Z_1$, $\mathcal{S}_1^l \perp\!\!\!\perp \mathcal{Z}_1^l$ and $\mathcal{S}_1^u \perp\!\!\!\perp \mathcal{Z}_1^u$.

  **Induction step**: Suppose $\mathcal{S}_i^u \perp\!\!\!\perp \mathcal{Z}_i^u$ and $\mathcal{S}_i^l \perp\!\!\!\perp \mathcal{Z}_i^l$, the analyst updates $Q_{i-1}(z \mid \mathbf{s})$ to $Q_i(z \mid \mathbf{s})$ with $\mathcal{S}_i^u \perp\!\!\!\perp \mathcal{Z}_i^u$ and $\mathcal{S}_i^l \perp\!\!\!\perp \mathcal{Z}_i^l$, makes a query on a label based on the prediction of $Q_i(z \mid \mathbf{s})$ and includes the (i+1)-th variable pair $(\mathbf{S}_{i+1}, Z_{i+1})$ to update the statistic. That immediately implies $\mathbf{S}_{i+1} \perp\!\!\!\perp Z_{i+1}$, $\mathcal{S}_{i+1}^u \perp\!\!\!\perp \mathcal{Z}_{i+1}^u$, and $\mathcal{S}_{i+1}^l \perp\!\!\!\perp \mathcal{Z}_{i+1}^l$.

  Combining the *base step* and the *induction step* leads to $S_i \perp\!\!\!\perp Z_i, \forall i \in [N_q]$ under $H_0$.

- **Proof for the second part**

  Suppose $((s, z)_i)_{i=1}^n$ is a sequence of realizations of $((\mathbf{S}, Z)_i)_{i=1}^n$ collected under $H_0$ and the proposed framework. We use $\phi$ to denote a class-one prior probability parameter, and hence $P(z_1, \cdots, z_n \mid \phi)$ is a likelihood function of $\phi$. Maximizing $P(z_1, \cdots, z_n \mid \phi)$ over the prior parameter $\phi$ leads to the solution $\phi^* = \frac{\sum_{i=1}^n z_i}{n}$. In other words, $P(z_1, \cdots, z_n \mid \phi^*) = \prod_{i=1}^n \hat{P}(z_i)$ is a maximized likelihood obtained from $(z_i)_{i=1}^n$, where $\phi^* = \hat{P}(Z = 1)$. We use $P(Z = 1)$ to denote the true prior-one probability under $H_0$,

and plugging $P(Z = 1)$ to $\phi$ leads to the true likelihood $\prod_i^n P(z_i)$ for $(z_i)_{i=1}^n$ under $H_0$. It is easy to see $\prod_{i=1}^n \hat{P}(z_i) \geq \prod_i^n P(z_i)$ thus $\prod_{i=1}^n \frac{\hat{P}(z_i)}{Q_i(z_i|\mathbf{s}_i)} \geq \prod_{i=1}^n \frac{P(z_i)}{Q_i(z_i|\mathbf{s}_i)}$ for any realization $(z_i)_{i=1}^n$ of $(Z_i)_{i=1}^n$ under $H_0$. As a result, we have $P_0 \left( \exists n \in [N_q] , W_n = \prod_{i=1}^n \frac{\hat{P}(Z_i)}{Q_{i-1}(Z_i|\mathbf{S}_i)} \leq \alpha \right) \leq P_0 \left( \exists n, \tilde{W}_n = \prod_{i=1}^n \frac{P(Z_i)}{Q_{i-1}(Z_i|\mathbf{S}_i)} \leq \alpha \right)$. Lastly, we prove $P_0 \left( \exists n, \tilde{W}_n = \prod_{i=1}^n \frac{P(Z_i)}{Q_{i-1}(Z_i|\mathbf{S}_i)} \leq \alpha \right) \leq \alpha$. We let $\tilde{W}'_n \equiv \frac{1}{\tilde{W}_n}$. Therefore, $\tilde{W}'_n \equiv \tilde{W}'_{n-1} \frac{Q_n(Z_n|\mathbf{S}_n)}{P(Z_n)}$ with $\tilde{W}'_0 \equiv 1$ for $n \in [N_q]$. The sequence $(\tilde{W}'_i)_{i=1}^n$ is a **non-negative martingale** under $H_0$ given

$$\cdot \mathbb{E}\left[ \tilde{W}'_n \middle| \tilde{W}'_1, \cdots, \tilde{W}'_{n-1} \right] \equiv \mathbb{E}\left[ \tilde{W}'_{n-1} \frac{Q_n(Z_n \mid \mathbf{S}_n)}{P(Z_n)} \middle| \tilde{W}'_1, \cdots, \tilde{W}'_{n-1} \right] \tag{23}$$

$$\equiv \tilde{W}'_{n-1} \mathbb{E}\left[ \frac{Q_n(Z_n \mid \mathbf{S}_n)}{P(Z_n)} \middle| \tilde{W}'_1, \cdots, \tilde{W}'_{n-1} \right] \tag{24}$$

$$= \tilde{W}'_{n-1} \mathbb{E}\left[ \sum_{z=0}^{1} P(Z_n = z) \frac{Q_{n-1}(Z_n = z \mid \mathbf{S}_n)}{P(Z_n = z)} \right] \tag{25}$$

$$= \tilde{W}'_{n-1} \tag{26}$$

Using Ville's maximal inequality in Theorem A.2 leads to the following: For any $\alpha > 0$, we have

$$P\left( \sup_{n\in[N_q]} \tilde{W}'_n > \frac{1}{\alpha} \right) \leq \frac{\alpha}{\mathbb{E}[\tilde{W}'_0]} = \alpha \tag{27}$$

$$\equiv P\left( \sup_{n\in[N_q]} \frac{1}{\tilde{W}_n} > \frac{1}{\alpha} \right) \leq \frac{\alpha}{\mathbb{E}\left[ \frac{1}{\tilde{W}_0} \right]} = \alpha \tag{28}$$

$$\equiv P\left( \inf_{n\in[N_q]} \tilde{W}_n \leq \alpha \right) \leq \alpha \tag{29}$$

Therefore, we have $P_0 \left( \exists n \in [N_q] , W_n = \prod_{i=1}^n \frac{\hat{P}(Z_i)}{Q_i(Z_i|\mathbf{S}_i)} \leq \alpha \right) \leq P_0 \left( \exists n \in [N_q] , \tilde{W}_n = \prod_{i=1}^n \frac{P(Z_i)}{Q_i(Z_i|\mathbf{S}_i)} \leq \alpha \right) \leq \alpha$.

$\square$

## B  Proof of Theorem 5.4

*Proof.* In the following, we formulate an optimization problem that seeks an arbitrary marginal distribution $g(\mathbf{s})$ to maximize the mutual information (MI) between $\mathbf{S}$ and $Z$, where $(\mathbf{S}, Z) \sim g(s)p(z \mid \mathbf{s})$. Solving this optimization problem leads to a consistent bimodal query (see Definition 5.2), asymptotically minimizing the test statistic in equation 2.

- **Constructing an optimization problem that maximizes MI**
  We write $g(\mathbf{s})$ to denote an arbitrary probability distribution of $\mathbf{s}$. Recall $P(z \mid \mathbf{s})$ and $p(\mathbf{s})$ that indicate the class probability given $\mathbf{s}$ and a marginal probability distribution of $\mathbf{s}$ for the two-sample testing problem on the analyst's hand; we write $g(\mathbf{s}, z) = g(\mathbf{s})P(z \mid \mathbf{s})$ and $G(z) = \int g(\mathbf{s}, z)\, d\mathbf{s}$ to denote the joint probability distribution and the class prior for a new two-sample testing problem with the original $p(\mathbf{s})$ replaced by $g(\mathbf{s})$. The mutual information (MI) that characterizes the new two-sample testing problem is as follows

$$\text{MI} = -\sum_{z=0}^{1} (G(z)) \log(G(z)) + \int \left( \sum_{z=0}^{1} P(z \mid \mathbf{s}) \log(P(z \mid \mathbf{s})) \right) g(\mathbf{s}) d\mathbf{s} \tag{30}$$

  We expand equation 30 and consider the following optimization problem,

$$\max_{g(\mathbf{s})} -\sum_{z=0}^{1} \left( \int p(z \mid \mathbf{s})g(\mathbf{s})d\mathbf{s} \right) \log\left( \int p(z \mid \mathbf{s})g(\mathbf{s})d\mathbf{s} \right) + \int \left( \sum_{z=0}^{1} P(z \mid \mathbf{s}) \log(P(z \mid \mathbf{s})) \right) g(\mathbf{s})d\mathbf{s} \tag{31}$$

In other words, equation 31 is seeking an $g(\mathbf{s})$ to maximize the MI of a new two-sample testing problem with $p(z \mid \mathbf{s})$ provided by the original two-sample testing problem. In what follows, we will see that solving 31 leads to a probability distribution in which a consistent bimodal query (see Definition 5.2) results, proving the asymptotic property in Theorem 5.4. Instead of directly solving equation 31, we fix $G(Z=0) = \int P(Z=0 \mid s)g(\mathbf{s})d\mathbf{s} = u$, and resort to finding the solution of the following,

$$\min_{g(\mathbf{s})} \quad -\int \left( \sum_{z=0}^{1} P(z \mid \mathbf{s}) \log \left( P\left(z \mid \mathbf{s}\right) \right) \right) g(\mathbf{s})d\mathbf{s} \tag{32}$$

$$\text{s.t.} \quad \int P(Z=0 \mid \mathbf{s})g(\mathbf{s})d\mathbf{s} = u, \tag{33}$$

$$\int g(\mathbf{s})d\mathbf{s} = 1, \tag{34}$$

$$g(\mathbf{s}) \geq 0, \forall s \in \mathcal{S}. \tag{35}$$

Then, we approximate equation 32 with a discrete version of the same by partitioning the sample space $\mathcal{S}$ into $L$ balls $\{B\left(\mathbf{s}_i, r\right)\}_{i=1}^{L}$; in addition, $L > 2$. Each $B\left(\mathbf{s}, r\right) \in \{B\left(\mathbf{s}_i, r\right)\}_{i=1}^{L}$ has a radius $r$ centering at $\mathbf{s}$ leading to an approximation $\hat{P}(Z=0|\mathbf{s}) = \int P(Z=0|\mathbf{s})p\left(\mathbf{s} \mid B(\mathbf{s}, r)\right)d\mathbf{s}$, and a probability mass function $G(\mathbf{s}) = \int_{\mathbf{s} \in B(\mathbf{s}, r)} g(\mathbf{s})d\mathbf{s}$. Hence, we approximate equation 32 by the following linear programming (LP):

$$\min_{G(\mathbf{s})} \quad \sum_{i=1}^{L} H_i(Z)G(\mathbf{s}_i) \tag{36}$$

$$\text{s.t.} \quad \sum_{i=1}^{L} \hat{P}(Z=0 \mid \mathbf{s}_i)G(\mathbf{s}_i) = u, \tag{37}$$

$$\sum_{i=1}^{L} G(\mathbf{s}_i) = 1, \tag{38}$$

$$G(\mathbf{s}_i) \geq 0, \forall i \in [L]. \tag{39}$$

where $H_i(Z) = -\sum_{z=0}^{1} \hat{P}(z \mid s_i) \log \left( \hat{P}(z \mid \mathbf{s}_i) \right), \forall i \in [L]$ indicates constant coefficients in the LP in equation 36.

- **Solving the optimization problem**
  The constraints in equation 37 and equation 38 construct a region of feasible solutions to the considered LP in equation 36; we write this region $U = \{\mathbf{s} \mid \mathbf{s} \text{ is non-negative and } \mathbf{s} \text{ satisfies equation 37 and equation 38.}\}$. In addition, we need to make one more definition of one kind of solution to the system of linear equations, which is well-known in linear algebra.

**Definition B.1.** (Basic solutions) Let $A\mathbf{x} = b$ be a system of linear equations. Let $\{\mathbf{x}_{j_1}, \cdots, \mathbf{x}_{j_k}\}$ be positive and other entries be zero in $\mathbf{x}$. Then, if the corresponding columns $A_{j_1}, \cdots, A_{j_k}$ are linearly independent, then $\mathbf{x}$ is a basic solution to the system.

Moreover, we will need to apply the following Theorems to derive the optimal feasible solution for the LP.

**Theorem B.2.** *If the feasible region of an LP is bounded, then at least one optimal solution occurs at a vertex of the corresponding polytope (or the feasible region).*

**Theorem B.3.** *Let $U$ be the feasible region of a linear program. Then, $\mathbf{x} \in U$ is a basic feasible solution if and only if $x$ is a vertex of $U$.*

Theorem B.2 and Theorem B.3 are well-known in LP; we refer interested readers to (Miller, 2007) for the elaboration on their proofs. Theorem B.2 and B.3 suggests one optimal solution of equation 36 is a vector

$(G\left(\mathbf{s}_1\right), \cdots, G\left(\mathbf{s}_L\right))$ with at most two non-zero entries. Herein, we write $G(\mathbf{s}_{q_0})$ and $G(\mathbf{s}_{q_1})$ to denote the two non-zero entries. That reduces the LP in equation 36 to the following:

$$\max_{q_0,q_1} \left( \left( \sum_{z=0}^{1} \hat{P}\left(z \mid \mathbf{s}_{q_0}\right) \log \hat{P}\left(z \mid \mathbf{s}_{q_0}\right) \right) G\left(\mathbf{s}_{q_0}\right) + \left( \sum_{z=0}^{1} \hat{P}\left(z \mid \mathbf{s}_{q_1}\right) \log \hat{P}\left(z \mid \mathbf{s}_{q_1}\right) \right) G\left(\mathbf{s}_{q_1}\right) \right) \tag{40}$$

$$\text{s.t.} \quad \hat{P}\left(Z = 0 \mid \mathbf{s}_{q_0}\right) G\left(\mathbf{s}_{q_0}\right) + \hat{P}\left(Z = 0 \mid \mathbf{s}_{q_1}\right) G\left(\mathbf{s}_{q_1}\right) = u, \tag{41}$$

$$G(\mathbf{s}_{q_0}) + G(\mathbf{s}_{q_1}) = 1, \tag{42}$$

$$G(\mathbf{s}_{q_0}) \geq 0, G(\mathbf{s}_{q_1}) \geq 0. \tag{43}$$

For the sake of simplifying the expressions in what follows, we write

$$T_0 = \hat{P}\left(Z = 0 \mid \mathbf{s}_{q_0}\right) \log \hat{P}\left(Z = 0 \mid \mathbf{s}_{q_0}\right) + \left(1 - \hat{P}\left(Z = 0 \mid \mathbf{s}_{q_0}\right)\right) \log \left(1 - \hat{P}\left(Z = 0 \mid \mathbf{s}_{q_0}\right)\right), \tag{44}$$

$$T_1 = \hat{P}\left(Z = 0 \mid \mathbf{s}_{q_1}\right) \log \hat{P}\left(Z = 0 \mid \mathbf{s}_{q_1}\right) + \left(1 - \hat{P}\left(Z = 0 \mid \mathbf{s}_{q_1}\right)\right) \log \left(1 - \hat{P}\left(Z = 0 \mid s_{q_1}\right)\right), \tag{45}$$

$$T_2 = \hat{P}\left(Z = 0 \mid \mathbf{s}_{q_1}\right) \log \hat{P}\left(Z = 0 \mid \mathbf{s}_{q_0}\right) + \left(1 - \hat{P}\left(Z = 0 \mid \mathbf{s}_{q_1}\right)\right) \log \left(1 - \hat{P}\left(Z = 0 \mid \mathbf{s}_{q_0}\right)\right), \tag{46}$$

$$T_3 = \hat{P}\left(Z = 0 \mid \mathbf{s}_{q_0}\right) \log \hat{P}\left(Z = 0 \mid \mathbf{s}_{q_1}\right) + \left(1 - \hat{P}\left(Z = 0 \mid \mathbf{s}_{q_0}\right)\right) \log \left(1 - \hat{P}\left(Z = 0 \mid s_{q_1}\right)\right). \tag{47}$$

Then, equation 40 is re-expressed by the following,

$$\max_{q_0,q_1} \frac{T_0 \left(u - \hat{P}\left(Z = 0 \mid s_{q_1}\right)\right)}{\hat{P}\left(Z = 0 \mid \mathbf{s}_{q_0}\right) - \hat{P}\left(Z = 0 \mid \mathbf{s}_{q_1}\right)} + \frac{T_1 \left(\hat{P}\left(Z = 0 \mid \mathbf{s}_{q_0}\right) - u\right)}{\hat{P}\left(Z = 0 \mid \mathbf{s}_{q_0}\right) - \hat{P}\left(Z = 0 \mid \mathbf{s}_{q_1}\right)} \tag{48}$$

$$\text{s.t.} \quad \hat{P}\left(Z = 0 \mid \mathbf{s}_{q_0}\right) - \hat{P}\left(Z = 0 \mid s_{q_1}\right) > 0, \tag{49}$$

$$\hat{P}\left(Z = 0 \mid \mathbf{s}_{q_1}\right) \leq u, \tag{50}$$

$$\hat{P}\left(Z = 0 \mid \mathbf{s}_{q_0}\right) \geq u. \tag{51}$$

equation 48 is an optimization problem that finds $\left\{\hat{P}\left(z \mid \mathbf{s}_{q_0}\right), \hat{P}\left(z \mid \mathbf{s}_{q_1}\right)\right\} \subset \{\hat{P}\left(z \mid \mathbf{s}_i\right)\}_{i=1}^{L}$ to maximize the objective function. Herein, we write

$$A = \frac{T_0}{\hat{P}\left(Z = 0 \mid \mathbf{s}_{q_0}\right) - \hat{P}\left(Z = 0 \mid \mathbf{s}_{q_1}\right)}, \tag{52}$$

$$B = u - \hat{P}\left(Z = 0 \mid \mathbf{s}_{q_1}\right), \tag{53}$$

$$C = \frac{T_1}{\hat{P}\left(Z = 0 \mid \mathbf{s}_{q_0}\right) - \hat{P}\left(Z = 0 \mid \mathbf{s}_{q_1}\right)}, \tag{54}$$

$$D = \hat{P}\left(Z = 0 \mid \mathbf{s}_{q_0}\right) - u. \tag{55}$$

Now, we analyze the derivatives of equation 48 by checking the partial derivatives of $A$, $B$, $C$ and $D$ with respect to $\hat{P}(Z = 0 \mid \mathbf{s}_{q_0})$ and $\hat{P}(Z = 0 \mid \mathbf{s}_{q_1})$:

$$\frac{\partial A}{\partial \hat{P}(Z = 0 \mid \mathbf{s}_{q_0})} = \frac{-T_2}{\left(\hat{P}(Z = 0 \mid \mathbf{s}_{q_0}) - \hat{P}(Z = 0 \mid \mathbf{s}_{q_1})\right)^2} > 0, \tag{56}$$

$$\frac{\partial A}{\partial \hat{P}(Z = 0 \mid \mathbf{s}_{q_1})} = \frac{T_0}{\left(\hat{P}(Z = 0 \mid \mathbf{s}_{q_0}) - \hat{P}(Z = 0 \mid \mathbf{s}_{q_1})\right)^2} < 0, \tag{57}$$

$$\frac{\partial B}{\partial \hat{P}(Z = 0 \mid \mathbf{s}_{q_1})} = -1, \tag{58}$$

$$\frac{\partial C}{\partial \hat{P}(Z = 0 \mid \mathbf{s}_{q_0})} = \frac{-T_1}{\left(\hat{P}(Z = 0 \mid \mathbf{s}_{q_0}) - \hat{P}(Z = 0 \mid \mathbf{s}_{q_1})\right)^2} > 0, \tag{59}$$

$$\frac{\partial C}{\partial \hat{P}(Z = 0 \mid \mathbf{s}_{q_1})} = \frac{T_3}{(\hat{P}(Z = 0 \mid \mathbf{s}_{q_0}) - \hat{P}(Z = 0 \mid \mathbf{s}_{q_1}))^2} < 0, \tag{60}$$

$$\frac{\partial D}{\partial \hat{P}(Z = 0 \mid \mathbf{s}_{q_0})} = 1. \tag{61}$$

Therefore, equation 48 is a function that monotonically increases with increasing $\hat{P}(Z = 0 \mid \mathbf{s}_{q_0})$ and decreasing $\hat{P}(Z = 0 \mid \mathbf{s}_{q_1})$, implying that the optimal solution to equation 36 has the following probability mass function $G^*$,

$$G^*(\mathbf{s}_{q_0}) = \frac{u - \hat{P}(Z = 0 \mid \mathbf{s}_{q_1})}{\hat{P}(Z = 0 \mid \mathbf{s}_{q_0}) - \hat{P}(Z = 0 \mid \mathbf{s}_{q_1})}, \mathbf{s}_{q_0} = \arg\max_{\mathbf{s}} \hat{P}(Z = 0 \mid \mathbf{s}), \tag{62}$$

$$G^*(\mathbf{s}_{q_1}) = \frac{\hat{P}(Z = 0 \mid s_0) - u}{\hat{P}(Z = 0 \mid \mathbf{s}_{q_0}) - \hat{P}(Z = 0 \mid \mathbf{s}_{q_1})}, \mathbf{s}_{q_1} = \arg\max_{s} \hat{P}(Z = 1 \mid \mathbf{s}) \tag{63}$$

$$G^*(\mathbf{s}) = 0, \forall \mathbf{s} \in \{\mathbf{s}_i\}_{i=1}^{L} \setminus \{\mathbf{s}_{q_0}, \mathbf{s}_{q_1}\}. \tag{64}$$

Recall that LP in equation 36 approximates the continuous optimization problem in equation 32 by partitioning the sample space $\mathcal{S}$ to $\{B(\mathbf{s}_i, r)\}_{i=1}^{L}$. Hence, by shrinking the radius $r$ infinitely close to zero, we get the optimal solution $p^*(\mathbf{s})$ of equation 32 as follows,

$$\frac{p^*(\mathbf{s}_{q_0})}{p^*(\mathbf{s}_{q_1})} = \frac{u - P(Z = 0 \mid \mathbf{s}_{q_1})}{P(Z = 0 \mid \mathbf{s}_{q_0}) - u}, \mathbf{s}_{q_0} = \arg\max_{\mathbf{s}} \hat{P}(Z = 0 \mid \mathbf{s}), \mathbf{s}_{q_1} = \arg\max_{\mathbf{s}} \hat{P}(Z = 1 \mid \mathbf{s}), \tag{65}$$

$$p^*(\mathbf{s}) = 0, \forall \mathbf{s} \in \mathcal{S} \setminus \{\mathbf{s}_{q_0}, \mathbf{s}_{q_1}\}. \tag{66}$$

Varying $u$ leads to the optimal solution with the same form that $p^*(\mathbf{s}) = 0, \forall s \in \mathcal{S} \setminus \{\mathbf{s}_{q_0}, \mathbf{s}_{q_1}\}$ and $p^*(\mathbf{s}_{q_0}) > 0, p^*(\mathbf{s}_{q_1}) > 0$, but different ratio $\frac{p^*(\mathbf{s}_{q_0})}{p^*(\mathbf{s}_{q_1})}$. Furthermore, there could exist a set $\mathcal{S}_{q_0} = \{\mathbf{s}_{q_0} \mid P(Z = 0 \mid \mathbf{s}_{q_0}) = \max_{\mathbf{s} \in \mathcal{S}} P(Z = 0 \mid \mathbf{s})\}$ with identical $P(z \mid \mathbf{s}_{q_0})$, and so does $\mathcal{S}_{q_1} = \{\mathbf{s}_{q_1} \mid P(Z = 1 \mid \mathbf{s}_{q_1}) = \max_{\mathbf{s} \in \mathcal{S}} P(Z = 1 \mid \mathbf{s})\}$ for the case of $\mathbf{s}_{q_1}$. Hence, the optimal solution to the original optimization problem in equation 31 has the following form

$$p^*(\mathbf{s}) = 0, \forall \mathbf{s} \in \mathcal{S} \setminus \left(\mathcal{S}_{q_0} \bigcup \mathcal{S}_{q_1}\right), \text{ and } p^*(\mathbf{s}) > 0, \forall \mathbf{s} \in \mathcal{S}_{q_0} \bigcup \mathcal{S}_{q_1}, \tag{67}$$

$$\mathcal{S}_{q_0} = \left\{\mathbf{s}_{q_0} \middle| P(Z = 0 \mid \mathbf{s}_{q_0}) = \max_{\mathbf{s} \in \mathcal{S}} P(Z = 0 \mid \mathbf{s})\right\}, \tag{68}$$

$$\mathcal{S}_{q_1} = \left\{\mathbf{s}_{q_1} \mid P\left(Z = 1 \mid \mathbf{s}_{q_1}\right) = \max_{\mathbf{s} \in \mathcal{S}} P(Z = 1 \mid \mathbf{s})\right\}. \tag{69}$$

Therefore, there exists a *consistent bimodal query* resulting in an asymptotic distribution of the labeled feature variables admitting $p^*(\mathbf{s})$ (equation 67 to equation 69) to maximize MI and hence minimize the negated MI with $P(z \mid \mathbf{s})$ provided by the original two-sample testing problem.

□

## C  Proof of Theorem 5.10

*Proof.* **Testing power of the baseline case**: As the baseline case randomly samples features from $\mathcal{S}_u$ and queries their labels, then the resulting variable pair $(\mathbf{S}_n, Z_n)$ collected by the analyst admits $p(\mathbf{s}, z), \forall n \in [N_q]$, in which $p(\mathbf{s}, z)$ is the joint distribution that characterizes the original two-sample testing problem. In addition, $Q(z \mid \mathbf{s})$ is initialized and stable, and the class-prior $P(Z = 0)$ is provided in the case study. Given the label budget $N_q$ and the significance level $\alpha$, we have the following inequalities for the testing power in the case study:

$$P_1\left(\exists n \in [N_q], W_n = \prod_{i=1}^{n} \frac{P(Z_i)}{Q(Z_i \mid \mathbf{S}_i)} \leq \alpha\right) \geq P_1\left(W_{N_q} = \prod_{i=1}^{N_q} \frac{P(Z_i)}{Q(Z_i \mid \mathbf{S}_i)} \leq \alpha\right), (\mathbf{S}_n, Z_n) \sim p(\mathbf{s}, z) \quad (70)$$

The inequality in equation 70 is derived from sequentially comparing $w_n$ with $\alpha, \forall n \in [N_q]$ leading to a higher testing power than only comparing $w_n$ with $\alpha$ at $n = N_q$. We subsequently convert RHS of equation 70 as follows,

$$P_1\left(W_{N_q} = \prod_{i=1}^{N_q} \frac{P(Z_i)}{Q_0(Z_i \mid \mathbf{S}_i)} \leq \alpha\right) = P_1\left(\frac{\log(W_{N_q})}{N_q} = \frac{\sum_{i=1}^{N_q} \log\left(\frac{P(Z_i)}{Q(Z_i \mid \mathbf{S}_i)}\right)}{N_q} \leq \frac{\log(\alpha)}{N_q}\right) \quad (71)$$

Since $\{(\mathbf{S_i}, Z_i)\}_{i=1}^{N_q}$ is an i.i.d. sequence, we skip $i$ in $(\mathbf{S_i}, Z_i)$ and analyze $\mathbb{E}\left[\frac{P(Z)}{Q(Z \mid \mathbf{S})}\right]$ and $\mathrm{Var}\left[\frac{P(Z)}{Q(Z \mid \mathbf{S})}\right]$ for $(\mathbf{S}, Z) \sim p(\mathbf{s}, z)$ in the following,

$$\mathbb{E}\left[\log\frac{P(Z)}{Q(Z \mid \mathbf{S})}\right] = \mathbb{E}\left[\log\frac{P(Z)}{P(Z \mid \mathbf{S})} + \log\frac{P(Z \mid \mathbf{S})}{Q(Z \mid \mathbf{S})}\right] \quad (72)$$

$$= -I(S; Z) + D_{\mathrm{KL}}(P(z \mid \mathbf{s}) \| Q(z \mid \mathbf{s})) \quad (73)$$

$$\leq -I(S; Z) + \sqrt{\epsilon_1}; \quad (74)$$

$$\mathrm{Var}\left[\frac{P(Z)}{Q(Z \mid \mathbf{S})}\right] = \mathrm{Var}\left[\log\frac{P(Z)}{P(Z \mid \mathbf{S})} + \log\frac{P(Z \mid S)}{Q(Z \mid \mathbf{S})}\right] \quad (75)$$

$$\leq \mathrm{Var}\left[\log\frac{P(Z)}{P(Z \mid \mathbf{S})}\right] + \mathrm{Var}\left[\log\frac{P(Z \mid \mathbf{S})}{Q(Z \mid \mathbf{S})}\right] + 2\sqrt{\mathrm{Var}\left[\log\frac{P(Z)}{P(Z \mid \mathbf{S})}\right]\mathrm{Var}\left[\log\frac{P(Z \mid \mathbf{S})}{Q(Z \mid \mathbf{S})}\right]} \quad (76)$$

$$\leq \sigma^2 + \epsilon_1 + 2\sigma\sqrt{\epsilon_1}. \quad (77)$$

The inequalities in equation 74 and equation 77 are results of the following facts: $\epsilon_1 = \max_{A \in \mathcal{P}} D_{\mathrm{KL}^2}(q(\mathbf{s}, z) \| p(\mathbf{s}, z) \mid A)$ and $\sigma^2 = \max\left\{\max_{A \in \mathcal{P}} \mathrm{Var}_{(\mathbf{S}, Z) \sim p(\mathbf{s}, z \mid A)} \bar{I}(\mathbf{S}; Z), \mathrm{Var}_{(\mathbf{S}, Z) \sim p(\mathbf{s}, z)} \bar{I}(\mathbf{S}; Z)\right\}$ over the partition $\mathcal{P} = \{A_1, \cdots, A_m\}$.

It is observed that, in equation 71, $\frac{\log(W_{N_q})}{N_q} = \frac{\sum_{i=1}^{N_q} \log\left(\frac{P(Z_i)}{Q(Z_i \mid \mathbf{S}_i)}\right)}{N_q}$ is a sample mean of $\left\{\log\frac{P(Z_i)}{Q(Z_i \mid \mathbf{S}_i)}\right\}_{i=1}^{N_q}$, hence we use the central limit theorem to approximate the distribution of $\frac{\log(W_{N_q})}{N_q}$ leading to the following,

$$P_1\left(\exists n \in [N_q], W_n = \prod_{i=1}^{n} \frac{P(Z_i)}{Q(Z_i \mid \mathbf{S}_i)} \leq \alpha\right) \geq P_1\left(W_{N_q} = \prod_{i=1}^{N_q} \frac{P(Z_i)}{Q(Z_i \mid \mathbf{S}_i)} \leq \alpha\right) \quad (78)$$

$$\approx \Phi\left(\frac{\frac{\log\alpha}{\sqrt{N_q}} + \sqrt{N_q}\left(I(\mathbf{S}; Z) - \sqrt{\epsilon_1}\right)}{\left(\sigma^2 + \sqrt{\epsilon_1} + 2\sqrt{\epsilon_1}\sigma\right)^{\frac{1}{2}}}\right). \quad (79)$$

**Testing power of the proposed framework in the case study:** The analyst selects a region $A^*$ from a partition $\mathcal{P} = \{A_i\}_{i=1}^m$, in which $A^*$ is predicted to have highest $I(\mathbf{S}; Z \mid A^*)$; then the analyst conducts the sequential testing with $(\mathbf{S}_n, Z_n)$ i.i.d. generated from $p(\mathbf{s}, z \mid A^*)$. We first quantify $I(\mathbf{S}; Z \mid A^*)$. Recall that the approximated MI $\left\{\hat{I}(\mathbf{S}; Z \mid A_i)\right\}_{i=1}^m$ used to find $A^* \in \mathcal{P}$ is provided in equation 16 in the case study; given Assumption 5.9, the discrepancy between true and approximate MI for any $A \in \mathcal{P}$ is as follows

$$I(\mathbf{S}; Z \mid A) - \hat{I}(\mathbf{S}; Z \mid A) = \mathbb{E}_{\mathbf{S} \sim p(\mathbf{s}|A)}\left[\mathbb{E}_{Z \sim Q(z|\mathbf{S})}[\log Q(Z \mid \mathbf{S})] - \mathbb{E}_{Z \sim P(z|\mathbf{S})}[\log P(Z \mid \mathbf{S})]\right] \tag{80}$$

Furthermore, given $\epsilon_2 = \max_{A \in \mathcal{P}} D_{\mathrm{KL}^2}(p(\mathbf{s}, z) \| q(\mathbf{s}, z) \mid A)$ over the partition $\mathcal{P} = \{A_1, \cdots, A_m\}$, we evaluate the upper bound of equation 80 for any $A \in \mathcal{P}$ in the following,

$$\mathbb{E}_{\mathbf{S} \sim p(\mathbf{s}|A)}\left[\mathbb{E}_{Z \sim Q(z|\mathbf{S})}[\log Q(Z \mid \mathbf{S})] - \mathbb{E}_{Z \sim P(z|\mathbf{S})}[\log P(Z \mid \mathbf{S})]\right] \tag{81}$$

$$\leq \mathbb{E}_{\mathbf{S} \sim p(\mathbf{s}|A)}\left[\mathbb{E}_{Z \sim Q(z|\mathbf{S})}[\log Q(Z \mid \mathbf{S})] - \mathbb{E}_{Z \sim Q(z|\mathbf{S})}[\log P(Z \mid \mathbf{S})]\right] \tag{82}$$

$$= D_{\mathrm{KL}}(Q(z \mid \mathbf{s}) \| P(z \mid \mathbf{s}) \mid A) \tag{83}$$

$$\leq \sqrt{\epsilon_2}. \tag{84}$$

Similarly, we evaluate the lower bound of equation 80 for any $A \in \mathcal{P}$ in the following,

$$\mathbb{E}_{\mathbf{S} \sim p(\mathbf{s}|A)}\left[\mathbb{E}_{Z \sim Q(z|\mathbf{S})}[\log Q(Z \mid \mathbf{S})] - \mathbb{E}_{Z \sim P(z|\mathbf{S})}[\log P(Z \mid \mathbf{S})]\right] \tag{85}$$

$$\geq \mathbb{E}_{\mathbf{S} \sim p(\mathbf{s}|A)}\left[\mathbb{E}_{Z \sim P(z|\mathbf{S})}[\log Q(Z \mid \mathbf{S})] - \mathbb{E}_{Z \sim Q(z|\mathbf{S})}[\log P(Z \mid \mathbf{S})]\right] \tag{86}$$

$$= - D_{\mathrm{KL}}(P(z \mid \mathbf{s}) \| Q(z \mid \mathbf{s}) \mid A) \tag{87}$$

$$\geq - \sqrt{\epsilon_1}. \tag{88}$$

Assumption 5.8 suggests that the maximum MI over $\mathcal{P}$ is $I(\mathbf{S}; Z) + \Delta$. Combining equation 84 and equation 88, we get the lower bound of $I(\mathbf{S}; Z \mid A^*)$ as follows,

$$I(\mathbf{S}; Z \mid A^*) \geq I(\mathbf{S}; Z) + \Delta - (\sqrt{\epsilon_1} + \sqrt{\epsilon_2}). \tag{89}$$

The analyst conducts the sequential testing in the selected $A^*$ with sample features randomly sampled from $A^* \bigcap \mathcal{S}_u$ and labeled, leading to the following testing power lower bound

$$P_1\left(\exists n \in [N_q], W_n = \prod_{i=1}^n \frac{P(Z_i)}{Q(Z_i \mid \mathbf{S}_i)} \leq \alpha\right) \geq P_1\left(W_{N_q} = \prod_{i=1}^{N_q} \frac{P(Z_i)}{Q(Z_i \mid \mathbf{S}_i)} \leq \alpha\right), (\mathbf{S}_n, Z_n) \sim p(\mathbf{s}, z \mid A^*). \tag{90}$$

The quantification of the RHS in equation 90 is identical to the one in the baseline case, except the sample space is constrained to $A^*$. Hence, we skip the derivation process and obtain the following result,

$$P_1\left(\exists n \in [N_q], W_n = \prod_{i=1}^n \frac{P(Z_i)}{Q(Z_i \mid \mathbf{S}_i)} \leq \alpha\right) \geq P_1\left(W_{N_q} = \prod_{i=1}^{N_q} \frac{P(Z_i)}{Q(Z_i \mid \mathbf{S}_i)} \leq \alpha\right) \tag{91}$$

$$\approx \Phi\left(\frac{\frac{\log \alpha}{\sqrt{N_q}} + \sqrt{N_q}\left(I(\mathbf{S}; Z) + \Delta - 2\sqrt{\epsilon_1} - \sqrt{\epsilon_2}\right)}{\left(\sigma^2 + \sqrt{\epsilon_1} + 2\sqrt{\epsilon_1}\sigma\right)^{\frac{1}{2}}}\right). \tag{92}$$

$\square$

