# OpenReview forum: "Active Sequential Two-Sample Testing"
_TMLR — Accepted by TMLR_

### Review · Reviewer_n3Ux · 2024-02-29

**Summary Of Contributions:**

The authors introduce active sequential two-sample hypothesis testing. In this setting, a learner has sample access to a distribution over unlabeled features and corresponding labels, and wishes to test whether the feature and label spaces are correlated. In other words given a joint distribution **S** over data and **Z** over labels, distinguish between:

$\forall s,z: P(z|s)=p(s)p(z)$ and $\exists s: P(z|s) \neq p(s)p(z)$

The authors study a natural pool-based active variant of this model in which the learner is assumed to have "pre-drawn" a large database of unlabeled samples $S_u \sim$ **S**, and may adaptively query for labels of points $s_u \in S_u$ drawn from $p(z|s_u)$. This is a natural setting for applications in e.g. digital health, where a practitioner might want to use `cheap’ indicators (raw data like speech, gait, age…) rather than costly bio-markers/imaging (labels). An active test in this setting allows for practitioners to test such a correlation using many fewer expensive label queries.

The authors introduce a natural sequential testing algorithm in which the practitioner always queries the sample $s_u \in S_u$ which she "believes" should maximize the conditional label probability $\Pr(Z=1|s)$ or $\Pr(Z=0|s)$, since these points contain the most information regarding correlation in the distribution. The practitioner’s `beliefs’ about such points are captured by a separate function $Q(z|s)$, which is pre-trained algorithm on a small initial sub-sample of data. The authors then track the statistic:

$$W_n = \prod \frac{P(z_i)}{Q_i(z_i|s_i)}$$

Where $(z_i,s_i)$ are the adaptively observed labeled samples and $Q_i$ is the belief function run on the prior data to this point.
The authors prove that regardless of choice of $Q$ and the dependencies introduced by active sampling, $W_n$ is still an “anytime-valid p-value”, that is it satisfies:

$$\Pr( \exists n: W_n \leq \alpha) \leq \alpha$$

Thus the practitioner can sequentially continue their test until they hit $W_n \leq \alpha$ while maintaining statistical guarantees.
The authors then show how the power of their framework is captured by the metal information between **Z** and **S**. In particular, in the asymptotic regime a normalized variant of their restriction converges to the mutual information. In the finite sample regime, the authors study a variant of their framework in which the practitioner has a partition of the unlabeled data, tests which subset of the partition has the highest mutual information, then performs the above sequential test. They show that if $Q$ is a good approximation for the true joint distribution, then the testing power of their active algorithm outperforms a standard `passive’ (non-adaptive) test by a factor of the mutual information.

Finally, the authors run several experiments, over simulated data, MNIST, and on an Alzheimer’s disease dataset vs the passive sequential baseline and show that both type I and type II error incurred by a fixed label budget are significantly lowered using the active approach.

**Audience:**

Yes

**Claims And Evidence:**

Yes

**Requested Changes:**

Some small comments:

$Q_n$ also needs to be defined more clearly and separately from the test statistic. It should not just be mentioned in the equation box for (2).

Since TMLR is not a statistics journal, I suggest adding a prelims section with basic statistics terms, e.g. test power. These may not be as well known in the ML community and aren’t otherwise introduced.

“parallel sequential testing baseline” is not defined in Thm 5.10

**Strengths And Weaknesses:**

**Strengths**

-The authors introduce a new, natural, and interesting setting combining ideas in active learning and statistics.

-The introduced method is "robust" in the sense that it remains statistically valid despite the dependence introduced by adaptivity and even if the initial $Q$ `belief’ function is wrong.

-Mutual information is a natural quantity to characterize the gain of an active learning problem in this setting. Both this and the above require some non-trivial work to show.

-While I am not qualified to review the experimental part of the paper, the methodology seems very successful in the simulated circumstances (beating baseline methods), including even on real medical data.

**Weaknesses**

-My only real complaint is that the `case study’ in section 5.4 where finite sample bounds are given (which is a bit more typical a setting for the machine learning literature) is not well motivated. In particular, it is very unclear what is going on with the partition of the unlabeled data set. Why is this here? Where is it coming from? Is the $m=1$ setting just the same as the prior setup, and this is a generalization? Please add some discussion on this.

-I was initially somewhat confused about the assumption that $Q$ is already a good estimator of the joint distribution. If this is the case, then naively one would think you could simply look at $Q$ and determine whether the distribution is independent. The authors could be more clear about the benefits of their framework even when the estimator is already good (namely, if I understand correctly, this gives a way of statistically verifying the correlation).

---

> ### Author Response · Authors · 2024-05-05
> **Thank you for your reviews! Please find our responses to your questions below.**
>
> **1. "My only real complaint is that the `case study’ in section 5.4 where finite sample bounds are given (which is a bit more typical setting for the machine learning literature) is not well motivated."**\
> Thank you for your insightful inquiries! The partitioning of the sample space $\mathcal{S}$  primarily facilitates mathematical analysis, allowing us to provide a closed form to quantify the advantages of the active scheme over the baseline. In particular, we can incorporate the error of selecting a non-max mutual information (MI) region into the testing power of the proposed test by partitioning $\mathcal{S}$.
>
> We are currently exploring an application in clinical trials where the partitioning becomes important: Given the heterogeneity of patients, treatment effects may be small. Therefore, clinicians partition patients based on covariates such as gender, age, and disease severity, aiming to identify groups most responsive to the treatment. By framing this clinical scenario as a two-sample test problem that tests whether treatment is effective within patient groups, our developed approach in the case study can be extended to be used in such clinical applications.
>
> **2. "I was initially somewhat confused about the assumption that is already a good estimator of the joint distribution. If this is the case, then naively one would think you could simply look at $Q$ and determine whether the distribution is independent. The authors could be more clear about the benefits of their framework even when the estimator is already good (namely, if I understand correctly, this gives a way of statistically verifying the correlation)."**\
> Thank you for your reassurance of our framework’s benefit that draws statistically significant conclusions by looking at the likelihood ratio statistic (Equation (2)). Indeed, even $P(z| \mathbf{s})$ is well approximated by $Q(z\mid\mathbf{s})$, simply looking at $Q$ can not make a reliable decision on whether to reject $H_0$. For instance, when $H_0$ is true,  looking at the imperfect estimator $Q$ without taking into account the generated data  $((\mathbf{s}, z)_i)$ could always lead to the rejection of $H_0$.
>
> **3. "Since TMLR is not a statistics journal, I suggest adding a prelims section with basic statistics terms, e.g., test power. These may not be as well known in the ML community and aren’t otherwise introduced."**\
> Thank you for your advice! We have revised the manuscript and added section 3.3, which extends the preliminary section.
>
> **4. “parallel sequential testing baseline” is not defined in Thm 5.10**\
> Thank you for bringing this to our attention.The parallel sequential testing baseline is initially introduced in the baseline case in the box under section 5.4.3. In order to reduce confusion, we have modified the term 'sequential testing baseline' to 'baseline test' in Theorem 5.10.

---

### Review · Reviewer_ZaZG · 2024-03-13

**Summary Of Contributions:**

This paper proposes a new active sequential procedure to perform a two-sample test, where sample features are inexpensive, but their group labels are costly. The proposed test is based on the likelihood ratio between the estimated marginal distribution and a probabilistic model trained using currently available data. Theoretically, it is shown that the false alarm probability can be controlled by providing a valid p-value for all time steps. As for the testing power, asymptotic and finite-sample analyses are provided to demonstrate the gain in the information compared to the baseline method. Empirically, experiments on synthetic, MNIST, and application-specific datasets validate those theoretical claims.

**Audience:**

Yes

**Claims And Evidence:**

Yes

**Requested Changes:**

I am curious about one thing. When you run out of the labeling budget, instead of only using the labeled samples to compute the likelihood ratio, can we use the well-trained model $Q_{z|S}$ to generate pseudo-labels for the remaining unlabeled data and compute the test statistics? Can you provide either theoretical or empirical evidence to show it will be better or worse? In general, will there be any benefit of using this model to generate pseudo-labels for the two-sample test?

In Table 3, it is not clear to me why the average number of label queries as a function of $\delta$ is not monotone. The maximum is achieved at $\delta =0.3$. Are there any possible explanations?


It should be Figure 2 instead of Table 2 on the first line on Page 11.

**Strengths And Weaknesses:**

Strength
-  The test procedure is simple and intuitive and works well in practice.
-  Theoretically, it is shown to have control over the false alarm probability, and the improvement over the testing power is justified via both asymptotic and finite sample analysis.

Weakness:
- The finite sample analysis has too many assumptions that are hard to verify in practice. I feel that given these conditions, it should be straightforward to prove the claim, but the challenging part is to verify the conditions. I feel that the formulation of this analysis is a bit contrived.
- The asymptotic analysis provides an intuitive way to understand the benefits of the proposed test in terms of the error exponent (i.e., the KL divergence between the null and alternative hypotheses), but it requires the labeled samples to go to infinite, which somehow contradicts to the idea of active learning to improve performance in finite sample regime.

---

> ### Author Response · Authors · 2024-05-05
> **Thank you for your constructive reviews! Please find our responses to your questions below.**
>
> **1. "The finite sample analysis has too many assumptions that are hard to verify in practice. I feel that given these conditions, it should be straightforward to prove the claim, but the challenging part is to verify the conditions. I feel that the formulation of this analysis is a bit contrived."**\
> Thank you for pointing this out. While they were labeled as assumptions in the manuscript, it would be more accurate to classify Assumption 5.7 (now Def 5.6 in the revised version) and Assumption 5.8 (now equation (15) in the revised version) as parameters that determine the testing power. We have identified that these two parameters correspond precisely to a divergence between two probability distributions and the relative entropy variance. To clarify this, we have added sections 5.4.1 and 5.4.2 to discuss the divergence and relative entropy variance. These two metrics are used to characterize the approximation error of $Q(z\mid\mathbf{s})$ (Assumption 5.7 in the original manuscript) and the irreducible Type II error (Assumption 5.8 in the original manuscript). As we see in the main result (Theorem 5.10), the finite-sample testing powers are determined by these metrics.
>
> Assumption 5.6 (now Assumption 5.8 in the revised version) embodies a practical consideration that there exist regions of interest (e.g., high MI regions) for active learning to leverage for improvement. If such regions are absent, active learning will yield results similar to those of the baseline (i.e., randomly querying labels) method. Lastly, Assumption 5.9 assumes the number of unqueried features is large. This assumption aligns with active learning's premise that the size of the sample features is big, but the label budget is small. Notably, a similar premise is also acknowledged in footnote 2 of [Hanneke, 2015].
>
> **2. "The asymptotic analysis provides an intuitive way to understand the benefits of the proposed test in terms of the error exponent (i.e., the KL divergence between the null and alternative hypotheses), but it requires the labeled samples to go to infinite, which somehow contradicts to the idea of active learning to improve performance in finite sample regime"**\
> You raise an important point. On the one hand, asymptotic analysis lies at the heart of developing valid statistical tests and has played a central role in the literature (see, e.g., [Henz, 1999]). As you note, this style of analysis gives us a strong understanding of how the underlying problem's characteristics affect the proposed methods' performance. To further make this work relevant to the data-efficient setting, we perform a finite-sample analysis in section 5.4 and quantify the benefit of the proposed framework over the baseline in the finite-sample case.
>
> **3. "When you run out of the labeling budget, instead of only using the labeled samples to compute the likelihood ratio, can we use the well-trained model to generate pseudo-labels for the remaining unlabeled data and compute the test statistics? Can you provide either theoretical or empirical evidence to show it will be better or worse? In general, will there be any benefit of using this model to generate pseudo-labels for the two-sample test"**\
> We can’t use the classifier outputs to perform the two-sample test after the label budget runs out. The reason is that the classifier’s probabilistic output is not a perfect estimate for $P(z\|s)$, and when $H_0$ ($\mathbf{S}$ and Z are dependent) is true, the generated pseudo-labels will be dependent on $\mathbf{S}$ resulting in $H_0$ **always** being rejected. We will add a line in the manuscript to clarify this.
>
> **4. "In Table 3, it is not clear to me why the average number of label queries as a function of $\delta$ is not monotone."**\
> Thank you for catching this. We apologize for the mistake in filling in the values of Table 3. We have corrected that in the revised vision. As the updated Table 3 shows, the average number of labels spent decreases with increasing $\delta$.
>
> **References**\
> Hanneke, Steve, and Liu Yang. "Minimax analysis of active learning." J. Mach. Learn. Res. 16.1 (2015): 3487-3602.\
> Henze, Norbert, and Mathew D. Penrose. "On the multivariate runs test." Annals of statistics (1999): 290-298.

---

### Review · Reviewer_KN7Q · 2024-04-20

**Summary Of Contributions:**

The paper introduces an Active Sequential Two-Sample Testing framework, addressing two-sample tests where sample features are inexpensive to collect, but their associated labels are costly. At a high level, the goal is to determine whether the distributions generating two samples are identical.  The algorithm has access to a fixed dataset of unlabeled covariates, and can sequentially query their label over time in an adaptive manner.  The test statistic uses a combination of $(i)$ the likelihood ratio, and $(ii)$ probabilistic classification model.  The classification model is adaptively updated and used to predict features and determine the new data to query the oracle for its label.  The hypothesis test procedure produces an anytime-valid $p$ value.  Additionally the authors investigate the algorithm's power through studying the mutual information between features and labels as well as accuracy of the classification procedure.

At a high level, the primary contributions for the paper can be summarized as follows:
- Extends the sequential hypothesis testing literature for two sample tests to consider the setting where the data labels need to be queried.  The authors additionally show that asymptotically the framework generates the largest mutual information between feature and label variables to help highlight its power.
- The authors conduct extensive experiments across synthetic, MNIST, and Alzheimer’s disease datasets to demonstrate the practical benefits of the framework. Results indicate significant reductions in Type II errors and more efficient use of label queries compared to traditional methods.

**Model:** There is a distribution $(S,Z)$ over feature label pairs.  The two sample testing problem consists of a null hypothesis $H_0$ that states that $p_{S | Z = 0} = p_{S | Z = 1}$ (i.e. the data label is independent of the features).  The algorithm is given a large set $S_0$ of sample features before performing the sequential test.  The statistician then sequentially queries a labelling oracle for the $z$ of each $s_u \in S_u$ subject to a maximum number of queries $N_q$.  After querying a new $z_n$, the analyst needs to decide whether to terminate the querying process and make a decision (i.e. accept or reject the null), or continue with the querying process.

The test statistic is then:
$$W_n = \prod_{i=1}^n \frac{\hat{P}(Z_i)}{Q_i(Z_i | S_i)}$$
and the procedure rejects the null whenever $w_n \leq \alpha$.  Note here that $\hat{P}(Z_i)$ is just the empirical distribution over the labels and $Q_i$ is the fit classification model.  When picking the next datapoint to query with a fair chance either pick the point which maximizes $Q(Z = 0 | s)$ or $Q(Z = 1 | s)$ where $s$ enumerates over the remaining unlabelled points in the dataset.

**Audience:**

Yes

**Broader Impact Concerns:**

No broader impact statement is necessary since the authors provide a theoretical statistical testing technique.

## Minor Comments
- Bolded statement on page one is great, really helps highlight the high level goal and contributions
- Figure 1 helps highlight the testing procedure, but would be helpful to introduce some of the notation (i.e. $z$ and $s$) earlier in the introduction to appreciate it more
- Problem statement box in 3.2 is really helpful
- Should define $P_0$ and $P_1$ formally at the bottom of page 4
- Would be useful to introduce potential classification methods $Q$ after they are first introduced in page 5 instead of postponing until later in the paper
- Some minor issues with capitalizing algorithm / equation throughout the paper
- Missing space after $w_n \leq \alpha$ on bottom of page 6
- Figure 2 vs Table 2 on top of page 11
- Bolded results in table 2 / 3 when the confidence intervals are overlapping feels a bit disingenuous, should modify them to only bold if there is statistical significance
- Issues with $=$ in the proofs

## Questions
- Surprised that Theorem 5.1 does not depend on the instantiation for the classification method $Q$.  Can you comment on why this is not directly needed to show the fact that the sequential hypothesis test is valid?

**Claims And Evidence:**

Yes

**Requested Changes:**

**Clarifying Proofs**: While I am not confident in this literature, I found several of the proofs difficult to follow and introduce notation that is not properly defined elsewhere in the paper.  For example, in Equation 19 on page 16, $\tilde{W}_n$ is introduced in the proof with no formal definition.

**Strengths And Weaknesses:**

## Strengths

The main strengths of the paper are:
- The paper extends the typical sequential hypothesis testing framework to include active querying, addressing a real-world problem where labels are expensive to collect. This novel approach not only improves efficiency in label usage but also increases the power of the testing framework under label constraints.
- The writing in the main paper is exceptionally clear and concise. The authors articulate the motivations, the theoretical advancements, and the introduction is very clear at highlighting the goal of the paper.
- The experimental results highlight the efficacy of the proposed framework across multiple datasets, including synthetic data and real-world applications like MNIST and Alzheimer's disease datasets.
- The paper provides substantial theoretical contributions, including the development of an anytime-valid p-value.

## Weaknesses

The main weaknesses of the paper are:

- While the paper cites relevant literature, it could better delineate its algorithmic  contributions relative to these works. Specifically, a more detailed discussion on how this framework advances beyond [Li et al (2022)].
- The proofs provided in the appendix are somewhat terse and difficult to follow.

---

> ### Author Response · Authors · 2024-05-04
> **We sincerely appreciate your reviews. Please find our responses to your questions below.**
>
> **1. "While the paper cites relevant literature, it could better delineate its algorithmic contributions relative to these works. Specifically, a more detailed discussion on how this framework advances beyond [Li et al (2022)]"**\
> Thank you for your advice! We have revised the manuscript and elaborated on their difference in the **related work**. In particular, our proposed framework is a sequential design that, when enough evidence is accumulated to draw a conclusion (i.e., reject $H_0$), our design automatically stops collecting labels before the label budget runs out. This is distinguished from [Li et al (2022)], whose batch design invariably exhausts the label budget.  In addition, our framework provides adaptability by allowing classifier updates to adjust the label query,  which is also not provided in [Li et al (2022)].
>
>  **2. "While I am not confident in this literature, I found several of the proofs difficult to follow and introduce notation that is not properly defined elsewhere in the paper. For example, in Equation 19 on page 16,  $\tilde{W}_n$ is introduced in the proof with no formal definition.**"\
> We appreciate your feedback regarding the proof of theorems in the Appendix. In response, we have expanded upon the explanations of our proofs in Sections A and B within the Appendix. We believe these additional details will assist reviewers and readers in understanding the logical progression of the proofs. $\tilde{ W }_n$ is a statistic in equation (2)
>
> with true class Prior $P(z)$ plugged in, resulting in  $\prod_{i = 1}^{n} \frac{ P(Z_i) }{ Q_{i}(Z_i\mid \mathbf{S}_i) }$. We have now formally introduced this before equation (22) (which is equation (19) in the original manuscript) in the revised version.
>
> **3. "Surprised that Theorem 5.1 does not depend on the instantiation for the classification method. Can you comment on why this is not directly needed to show the fact that the sequential hypothesis test is valid?**"\
> This stems from the fact that when $H_0$ is true ($Z$ and $\mathbf{S}$ are independent), we have $\mathbb{E}_{(\mathbf{S}, Z)\sim p(\mathbf{s})P(z)}\left[\frac{Q(Z\mid\mathbf{S})}{P(Z)}\right]=1$, regardless of $Q$.  Consequently,  the random sequence $\left(\frac{1}{\tilde{W}_i}\right)$ is a martingale under $H_0$, regardless of $Q$. Therefore, we can upper-bound the tail probability of the martingale, resulting in Theorem 5.1.  The detailed proof is provided in equations (23) to (29); similar results can also be found in proposition 1 of [Lhéritier, 2018]. This “classifier-invariance” is a key insight in our results, and we will be sure to draw the readers’ attention to this.
>
> **4. "Bolded results in table 2 / 3 when the confidence intervals are overlapping feels a bit disingenuous, should modify them to only bold if there is statistical significance"**\
> Thank you for your advice! We have run additional two-sample t-tests to assess the mean difference of label query numbers across 200 runs in synthetic, MNIST, and ANDI experiments.  Remarkably, all resulting $p$-value, truncated to the last six decimal places, equate to zero. As $p<0.05$ in the t-test implies a statistical difference between the mean, it is fair to say the proposed framework spent fewer labels than the baseline across all parameter sets in our experiments.
>
> **References**\
> Lhéritier, Alix, and Frédéric Cazals. "A sequential non-parametric multivariate two-sample test." IEEE Transactions on Information Theory 64.5 (2018): 3361-3370.

---

### Author Response · Authors · 2024-05-04
**General Response**

We extend our gratitude for your inspiring feedback and constructive critiques. In response to your valuable input, we have revised the manuscript accordingly. The revised sections are now highlighted in blue within the manuscript. Furthermore, we have restructured section 5.4, which analyzes the finite-sample testing power, highlighting that the testing power is governed by a divergence between two probability distributions and the relative entropy variance.  Lastly, we have individually addressed your questions, providing responses separately below.

---

### Decision · Action_Editor_2N5K · 2024-05-28

**Recommendation:** Accept as is

**Comment:**

The claims of the paper appear correct and the results are interesting.

**Audience:**

Audience will be the statistical part of the ml community -- for example, audience of the AISTATS conference.

**Claims And Evidence:**

This paper considers the two-sample testing problem in a new setting where the sample measurements (or sample features) are inexpensive to access, but their group memberships (or labels) are costly. All reviewers agree that the results are correct and interesting and this is an easy accept.